# Facile Purification and Use of Tobamoviral Nanocarriers for Antibody-Mediated Display of a Two-Enzyme System

**DOI:** 10.3390/v15091951

**Published:** 2023-09-19

**Authors:** Tim Wendlandt, Claudia Koch, Beate Britz, Anke Liedek, Nora Schmidt, Stefan Werner, Yuri Gleba, Farnoosh Vahidpour, Melanie Welden, Arshak Poghossian, Michael J. Schöning, Fabian J. Eber, Holger Jeske, Christina Wege

**Affiliations:** 1Institute of Biomaterials and Biomolecular Systems, Molecular and Synthetic Plant Virology, University of Stuttgart, Pfaffenwaldring 57, 70569 Stuttgart, Germany; tim.wendlandt@bio.uni-stuttgart.de (T.W.); dr-claudiakoch@web.de (C.K.); nora.schmidt@helmholtz-hiri.de (N.S.);; 2Nambawan Biotech GmbH/Now at Icon Genetics GmbH, Weinbergweg 22, 06120 Halle, Germany; werner@icongenetics.de; 3Nomad Bioscience GmbH, Weinbergweg 22, 06120 Halle, Germany; gleba@nomadbioscience.com; 4Institute of Nano- and Biotechnologies, Aachen University of Applied Sciences, 52428 Jülich, Germany; vahidpour@fh-aachen.de (F.V.); m.welden@fh-aachen.de (M.W.); schoening@fh-aachen.de (M.J.S.); 5MicroNanoBio, Liebigstrasse 4, 40479 Düsseldorf, Germany; 6Institute of Biological Information Processing (IBI-3), Forschungszentrum Jülich GmbH, 52425 Jülich, Germany; 7Department of Mechanical and Process Engineering, Offenburg University of Applied Sciences, 77652 Offenburg, Germany; fabian.eber@hs-offenburg.de

**Keywords:** turnip vein clearing virus (TVCV), tobacco mosaic virus (TMV), tobamovirus, enzyme nanocarrier, protein A, antibody, enzyme cascade, glucose oxidase (GOx), horseradish peroxidase (HRP), biosensor

## Abstract

Immunosorbent turnip vein clearing virus (TVCV) particles displaying the IgG-binding domains D and E of *Staphylococcus aureus* protein A (PA) on every coat protein (CP) subunit (TVCV_PA_) were purified from plants via optimized and new protocols. The latter used polyethylene glycol (PEG) raw precipitates, from which virions were selectively re-solubilized in reverse PEG concentration gradients. This procedure improved the integrity of both TVCV_PA_ and the wild-type subgroup 3 tobamovirus. TVCV_PA_ could be loaded with more than 500 IgGs per virion, which mediated the immunocapture of fluorescent dyes, GFP, and active enzymes. Bi-enzyme ensembles of cooperating glucose oxidase and horseradish peroxidase were tethered together on the TVCV_PA_ carriers via a single antibody type, with one enzyme conjugated chemically to its Fc region, and the other one bound as a target, yielding synthetic multi-enzyme complexes. In microtiter plates, the TVCV_PA_-displayed sugar-sensing system possessed a considerably increased reusability upon repeated testing, compared to the IgG-bound enzyme pair in the absence of the virus. A high coverage of the viral adapters was also achieved on Ta_2_O_5_ sensor chip surfaces coated with a polyelectrolyte interlayer, as a prerequisite for durable TVCV_PA_-assisted electrochemical biosensing via modularly IgG-assembled sensor enzymes.

## 1. Introduction

Engineered, elongated plant RNA viruses are attractive multivalent carrier scaffolds for applications in nanotechnology and biomedicine, due to their dimensions and shape defined by the encapsidated genome, and a surface exposing repetitive docking sites for functional molecules with spacing on the nanometer scale on a single type of coat protein (CP) subunits [1,2,3,4,5,6,7,8,9,10,11,12,13]. The high aspect ratio of rod-shaped and flexible filamentous viruses is advantageous for addressing tumors and atherosclerotic plaques via the bloodstream and cellular uptake of effector-loaded particles in vivo [13,14]. It also allows their assembly into dense networks on solid substrates and in solution, as well as retention in hydrogels, to serve as 2D- and 3D-immobilization platforms for various molecule species or cultivated cells [15,16,17,18]. Key players in this context are tobamoviruses such as tobacco mosaic virus (TMV), potex-, and potyviruses including potato virus X (PVX), and turnip mosaic virus (TuMV) [18,19,20,21,22,23,24,25,26,27,28,29,30,31,32,33,34].

Among these, TMV with its natural dimensions of 300 nm × 18 nm, an inner channel of 4 nm diameter that is also amenable to cargo loading, and more than 2100 CP subunits has been introduced into a large variety of applications [35]. Its easy production, physical and chemical stability, and options for generating TMV-derived designer particles of altered shapes offer versatile prospects for multi-functionalization with, e.g., peptides, enzymes, drugs, and/or synthetic and inorganic compounds, and corresponding uses [31]. Increasing numbers of studies have employed TMV-based adapters for the display of enzymes, and for installing enzymatic and other types of bioreceptors in different types of sensors [20,25,36,37,38,39,40,41,42,43,44,45,46,47,48,49,50,51]. TMV, however, does not tolerate extensive genetic modifications. Most applications of plant-harvested particles therefore make use of the wild-type (wt) TMV or variants with only moderate genetic CP alterations, from single up to about 25 amino acids (aa) replaced or fused [35], often in combination with costly conjugation chemistry. Genetically encoded displays of the 125 aa fluorescent reporter iLOV [52] have only been achieved via a sophisticated construct containing the foot-and-mouth disease virus (FMDV) 2A ‘ribosome skip’ sequence, limiting the portion of extended CPs co-translationally [53]. Enzymatic (self-)ligation of peptide- or coenzyme-A-equipped CPs to compatibly tagged proteins of interest is a promising alternative for installing functions but has been evaluated only exemplarily so far for TMV, with variable success [25,51]. 

Searching for versatile, reliable coupling strategies for enzymes to TMV-like particles, we went back to an unexpectedly extensive, genetically encoded surface modification achieved for a distinct tobamovirus more than 15 years ago: the display of IgG-affine 133 aa domains (D and E) of *Staphylococcus aureus* protein A (PA) on every CP subunit of turnip vein clearing virus (TVCV) [54,55,56], resulting in plant-infectious TVCV_PA_ with more than 2000 copies of the functional protein A segment on its capsid [57]. In the original study and recent consecutive work [58], the PA domain-fashioned virus-based immunosorbent nanoparticles (VINs) were developed for the efficient capture of antibodies and therapeutic antibody-fusion proteins. They allowed their convenient enrichment not only directly from *Nicotiana benthamiana* extracts but also after entrapment in a robust silica-based purification matrix [59]. 

We now seek to investigate the use of TVCV_PA_ as a bionanocarrier for enzymes installed via a surface-exposed antibody layer, which is of great interest especially for biosensor applications [15,60]. To this end, we introduce a rapid purification process newly established for TVCV_PA_ and TVCV, which overcomes previous quality constraints and might ease and speed up VIN isolation for routine uses. It is based on the concept of reverse polyethylene glycol (PEG) solubility gradients for a selective re-solubilization of virus–PEG precipitates and has been adapted to plant raw homogenates. We have re-investigated the size of the two vector-derived TVCV variants accumulated in plants systemically, in the case of TVCV_PA_ after the recombination of co-agroinfiltrated deconstructed parental pro-vector modules described earlier [57,58]. This resolved previous inconsistencies between the lengths reported for the TVCV_PA_ particles [57,58] and the underlying original TVCV strain [54,55,56]. The main goal of this study was to test whether TVCV_PA_ particles could be decorated with two enzymes of a cascade of glucose oxidase (GOx) and horseradish peroxidase (HRP) simply via bioaffinity capture, to allow the detection of glucose. The work builds on our previous findings on TMV that have revealed its benefits as a nanocarrier for sensor enzymes in microtiter plate formats [46,50]. As in our more recent work, TMV adapters also improved the performance of various label-free electrochemical biosensors; we additionally investigated if TVCV_PA_ VINs can be applied for coating sensor chip surfaces via procedures optimized for TMV.

In view of the further potential of TVCV_PA_ in new materials and biohybrid devices, some fundamentals and properties are briefly related to those of TMV. Both *Turnip vein clearing virus* and *Tobacco mosaic virus* are species in the genus *Tobamovirus* (family *Virgaviridae*), with TMV as the eponymous type member. The helical, nanotubular ribonucleoprotein particles of the viruses in more than 35 tobamovirus species all contain a single CP type and a positive-sense ssRNA of 6.3 to 6.5 kbases with similar genetic organization [61]. Tobamoviruses have been classified into at least three subgroups co-evolved with their distinct primary host plant taxa [62,63,64,65]. Obvious differences lie in the position of their origin of assembly (OAs) sequences in the RNA, and the separation or partial overlap of the movement protein (MP) and CP open reading frames (ORFs) [65,66]. TMV belongs to the subgroup 1 viruses, most frequently isolated from *Solanaceae.* Their OAs is located within the MP ORF that is separated from that of the CP by 2–5 nucleotides (nts). Further well-known members in this group are, e.g., tomato mosaic virus (ToMV) and tobacco mild green mosaic virus (TMGMV) [66]. Turnip vein clearing virus (TVCV) is a member of subgroup 3 viruses, which were isolated mainly from *Brassicaceae* and *Plantaginaceae*. In addition to its use as a bionanoparticle backbone, TVCV is interesting for fundamental research as it can infect *Arabidopsis thaliana* [67]. Its CP and MP ORFs overlap by 77 nts, with the OAs inside the latter as for TMV [68,69]. 

The evolutionary distance from TMV thus accounts for a slightly shorter RNA of TVCV (6311 nts for the underlying TVCV reference strain OSU [54,56] versus 6395 nts for TMV vulgare [70]), resulting in a predicted TVCV particle length of 296 nm, a few nm below that of TMV (300 nm). This is in agreement with the main lengths of 275–300 nm determined electron-microscopically for natural TVCV [55]. However, after the expression of TVCV, as well as TVCV_PA_ (with 6755 nts/317 nm, considering the elongated CP gene) from pro-vectors in planta, deviating dominant lengths of 200–220 nm were found for the purified particles [57]. 

TMV CP (158 aa following N-terminal M removal/17.49 kDa neglecting acetylation) and TVCV CP (156 aa after M removal/17.47 kDa) are predicted to have a 49% aa identity and highly similar folds (Figure 1). Correspondingly, virus particles are indistinguishable from each other via routine transmission electron microscopy (TEM), and minor differences between the two CPs’ calculated isoelectric points (pIs of TVCV CP: 5.04, TMV: 5.08, according to the Isoelectric Point Calculator 2.0 (IPC 2.0) [71], seem irrelevant in folded conformation and for standard purification or functionalization protocols. However, the PA domain-extended CP of TVCV_PA_ (33.05 kDa) has a calculated more negative pI of 4.82, with that of the C-terminal 148 aa fusion fragment alone of 4.73 (15 aa (G_4_S)_3_ linker + 133 aa PA domains E/D). It is thus expected to affect the viral surface charge considerably, with potential consequences for virus isolation, immobilization, and use in hybrid materials. A further important discrepancy is the position and orientation of the CPs’ lysines, which represent potential chemical conjugation sites. Both TVCV CP and TMV CP each have two lysines, which are, though, located at different positions in their sequences and thus are oriented in distinct directions in the assembled state (Figure 1). In contrast, the position of a natural cysteine potentially accessible at only one end of the virus particles is conserved in both viruses and might be suitable for the 5′ terminal coupling of TVCV rods in future work in an analogy to TMV [72].

## 2. Material and Methods

### 2.1. Plant Infection

#### 2.1.1. Delivery of Vector-Encoded Viral Constructs via Agroinfiltration

Systemically infectious TVCV or TVCV_PA_ were released in *N. benthamiana* DOMIN leaves from infiltrated agrobacterial vectors, as described [57,75,76]. The vector for TVCV inoculation encoded a completely assembled viral wt genome (with its cDNA interrupted by introns and cryptic splice sites removed: TVCV_WT_, pICH17501). TVCV_PA_ was delivered via two deconstructed viral pro-vector modules (pICH20701 and pICH21767) co-infiltrated in a mixture of both agrobacterium clones with a third providing bacteriophage ϕC31 integrase (pICH14011), which recombines the two virus-encoding vector portions in planta. The viral module in pICH20701 comprises the 5′ terminal partial TVCV genome including its CP ORF without stop codon, followed by the coding sequence for a 15 aa linker (TVCV-CP-[G_4_S]_3_) and a ϕC31 AttP recombination site. The 3′ module in pICH21767 encodes downstream the ϕC31 AttB recombination site the *S. aureus* protein A domains D+E and the viral 3′-NTR (non-translated region) [for details, see 57 and references therein]. *Agrobacterium tumefaciens* (GV3101::pMP90) carrying one of the TVCV_WT/PA_ (pro-) vectors or the integrase expression construct, respectively, were cultivated in YEB medium supplemented with appropriate antibiotics (Rifampicin [final concentration (f.c.) 25 µg mL^−1^], Gentamicin [f.c. 20 µg mL^−1^] for all plasmids, plus Kanamycin [f.c. 50 µg mL^−1^] for pICH17501 or plus Carbenicillin [f.c. 100 µg mL^−1^] for pICH20701, pICH21767, pICH14011) at 28 °C with shaking at 200 rpm overnight (ON). These cultures were diluted 1:200 in infiltration buffer (3.3 mM MES, 10 mM MgCl_2_, 100 µM acetosyringone) and infiltrated by a syringe without needle into two to three fully expanded upper leaves (each 200 µL bacteria suspension) of *N. benthamiana* in the five-to-six leaf stage. In initial experiments, diluted agrobacteria suspensions with acetosyringone were kept in the dark for 2–4 h prior to infiltration to ensure efficient T-DNA transfer, which came out to be unnecessary for routine applications. To produce TVCV_PA_, the clones pICH20701, pICH21767, and pICH14011 were combined in equal ratios; for TVCV_WT_, clone pICH17501 was agroinfiltrated alone. Plants treated with infiltration buffer (mock) served as negative controls.

#### 2.1.2. Mechanical Inoculation of Plants with Leaf Extracts or Purified Virus

Infected leaf samples (≈125 mg) were ground in 400 µL 10 mM sodium potassium phosphate (SPP) buffer pH 7.0. *N. tabacum* L. ‘Samsun’ nn or *N. benthamiana* DOMIN plants in the three-to-four-leaf stage were inoculated mechanically with 80 µL or 40 µL, respectively, of this homogenate per leaf onto the carborundum-dusted lamina of two to three upper leaves, and sprayed with water. Alternatively, purified virus particles were diluted in 100 µL 10 mM SPP pH 7.0 (final concentration [f.c.]: 0.5 µg/µL) and inoculated as above. Mock plants were treated with buffer. TMV vulgare applied in control experiments was inoculated mechanically to *N. tabacum* ‘Samsun’ nn or to *N. benthamiana* with leaf homogenates from plants infected via a plasmid clone [77], as described [78].

#### 2.1.3. Plant Harvest and Virus Immunodetection on Tissue Blots

Leaves from TVCV_WT_-infected, symptomatic plants were harvested 11–13 days post inoculation (dpi), from TVCV_PA_-infected plants 18–20 dpi, and stored at −20 °C or processed further. Tissue-print immunodetection identified TVCV-infected plants unequivocally [79]. Up to four randomly chosen leaves or leaf petioles per plant were pressed onto a nitrocellulose membrane briefly, which was then air-dried. One microgram of purified TMV, and/or TVCV_WT_, or TVCV_PA_ in 10 mM SPP was spotted onto the membrane as a positive control. The membrane was blocked with 1% (*w*/*v*) BSA in Tris-buffered saline (50 mM Tris-HCl, 150 mM NaCl pH 7.3) with 0.1% Tween-20 (TBS-T) for 30 min, followed by three washing steps with TBS-T. To detect TVCV, the membranes were incubated with rabbit anti-TMV antibodies (anti-TMV, IgG fraction, 240 µg/mL, diluted 1:1000 in TBS; provided by K.-W. Mundry and E. Hornung, University of Stuttgart) either at 4 °C overnight or at room temperature (RT) for 2–3 h, cross-reacting due to the high similarity of TMV and TVCV. Following washing (3 × 5 min in TBS-T), the blot was developed via incubation with alkaline phosphatase-(AP)-conjugated anti-rabbit IgG antibodies (goat anti-rabbit IgG-AP IgG, diluted 1:3000 in blocking solution; Rockland Immunochemicals, Limerick, PA, USA) at RT for 45 min, washing (6 × 5 min in TBS-T), equilibration in AP-reaction buffer (100 mM Tris-HCl pH 9.5, 100 mM NaCl, 5 mM MgCl_2_), and incubation with NBT/BCIP substrate solution, resulting in AP-mediated violet stain precipitation [80]. TVCV_PA_ particles were either detected equally or with a secondary antibody from rabbit (rabbit anti-goat IgG-AP) alone, which bound directly to the PA-domains prior to AP-mediated detection as above.

#### 2.1.4. Virus Isolation via Stepwise Enrichment: Butanol Extraction, Two-Fold PEG Precipitation, and Ultracentrifugation

TVCV_WT_ was isolated from fresh or frozen *N. benthamiana* leaves via a standard protocol for TMV purification from pre-cleared plant homogenates as modified by [81], based on [82], but at 4 °C. In brief, after disruption of plant materials in cold 0.5 M SPP pH 7.0 (corresponding to ≈7% initial Na_2_HPO_4/_KH_2_PO_4_ salt input with ≈2% monovalent cations) in a Waring blender, the homogenates were subjected to butanol extraction and two-fold PEG precipitation of virions first via a dropwise addition of 4% (*w*/*v*) PEG-6000 (f.c) /1% (*w*/*v*) NaCl (f.c.) from a 4-fold stock solution, collection by centrifugation, resuspension in 10 mM SPP pH 7.0 (0.16% salt) and a second precipitation via the addition of 4% PEG/4% NaCl (f.c.) as before. TVCV particles were further purified via differential ultracentrifugation (UC) of the resuspended pellet at 120,000× *g*, as described for TMV [83]. For TVCV_PA_, these conditions, as well as a related protocol applied for its isolation before (using 4% (*w*/*v*) PEG-6000 with ≈2.6% and 1.2% (*w*/*v*) f.c. total salt in the first and second precipitation, respectively) [57], resulted in lower yield and purity, as virion precipitation was incomplete. Hence, the above protocol was altered to keep the supernatant of the first PEG precipitate as a TVCV_PA_ source, re-extract it a second time with butanol, and precipitate virus particles via the addition of 4% PEG/4% NaCl (f.c.). Subsequent steps (transfer into SPP buffer and UC) followed the above procedure again. Virus-containing solutions were stored in 10 mM SPP buffer pH 7.0 to 7.2 supplemented with 1 mM phenylmethylsulfonyl fluoride (PMSF) and 2 mM ethylenediamine tetraacetic acid (EDTA) [57]. Particle concentrations were inferred from UV spectra (Nanodrop ND-1000 spectrophotometer; Peqlab, Erlangen, Germany) at 260 nm, using an extinction coefficient of 3 mg/mL∙cm as determined for TMV [84]. For control purposes, TMV was extracted from *N. tabacum* or *N. benthamiana* as described [83].

#### 2.1.5. Virus Isolation via Precipitates from Cleared Crude Extracts: Selective Inverse PEG-Sucrose Solubility Gradients

In the search for alternatives to laborious and time-consuming standard procedures for virus isolation, a contrarian approach based on selective re-solubilization of TVCV_PA_ from raw precipitates of plant homogenates was evaluated based on [85]. Inverse PEG-6000 gradients were generated in ultracentrifuge (UC) tubes by allowing six stacked liquid layers to diffuse into each other. From top to bottom, each layer contained a decreasing PEG-6000 concentration, and an increasing sucrose concentration as follows. Layers of 1.5 mL volume (in 50 mM SPP pH 7.2) each contained 5, 10, 15, 20, 25, 30% (*w*/*v*) sucrose and 5.5, 3.0, 2.0, 1.0, 0.5, 0.0% PEG-6000 (TVCV_WT_), or 8.0, 7.5, 7.0, 6.5, 6.0, 0.0% PEG-6000 (TVCV_PA_). These were stacked into 13.2 mL UC tubes (Ultra-Clear, Beckman Coulter) for SW 41 Ti rotors (Beckman Coulter, Krefeld, Germany) by dispensing each aliquot to the tubes’ bottom via an extra-long syringe needle, underlaying the preceding layer. The sharply separated layers were allowed to form a gradient by diffusion overnight (4 °C).

For each SW 41 Ti tube to be used later in the process, 8–10 g symptomatic leaves were ground in liquid nitrogen, suspended in 15 mL SPP buffer (500 mM, pH 7.2) containing 1% 2-mercaptoethanol, filtered through three layers of Miracloth (Millipore-Merck KGaA, Darmstadt, Germany) and clarified by centrifugation (4 °C, 10 min, 3000× *g* [RCF_max_] = 5000 rpm; SS-34 rotor). The resulting cleared crude extract was mixed with an equal volume of a PEG solution (50 mM SPP pH 7.2, 2% NaCl, 11% PEG-6000 for TVCV_WT_, 16% PEG-6000 for TVCV_PA_) and roller-mixed for 2 h at 4 °C. Subsequently, the PEG precipitate was sedimented (4 °C, 15 min, 10,000× *g* [RCF_max_] = 9150 rpm; SS-34 rotor,), resuspended in 1.5 mL of the pellet’s supernatant and laid onto prefabricated inverse PEG–sucrose solubility gradients. Tubes were centrifuged (12 °C, 30 min, ≈24,600× *g* [RCF_max_] = 12,000 rpm; SW 41 Ti rotor). For analytical purposes, the tubes’ contents were fractionated by pipetting equal volumes from top to bottom. Fractions were analyzed via SDS-PAGE (see below) for the presence of TVCV CP and contaminants. Alternatively, visible virus bands were sucked into a syringe by puncturing the tube with the needle. Samples of appropriate purity and concentration were pooled and washed via repetitive ultrafiltration in centrifugal devices five times (Amicon Ultra centrifugal filter units Ultra-4, molecular weight cut-off (MWCO): 100 kDa; Millipore-Merck KGaA, Darmstadt, Germany), restocking the ultrafiltration units for each run with 25 mM 3-(N-morpholino)propanesulfonic acid (MOPS) buffer pH 7.0, 5 mM EDTA.

### 2.2. Characterization of Virus Preparations

#### 2.2.1. Gel Electrophoresis

For denaturing polyacrylamide gel electrophoresis (PAGE), 1–1.5 µg virus particles were mixed with loading buffer (f.c. of 50 mM Tris-HCl pH 6.8, 2% (*w*/*v*) SDS, 0.1% (*w*/*v*) bromophenol blue, 10% (*v*/*v*) glycerol, 100 mM dithiothreitol), heated for 5 min at 95 °C, and proteins were separated in 12% SDS-PA gels [86]. Gels were stained with 0.5% (*w*/*v*) Coomassie Brilliant Blue R250 or G250 (Serva Electrophoresis, Heidelberg, Germany) according to standard protocols [80]. For native separation in agarose gels, 10 μg virus particles were loaded to a 0.8% agarose gel in 50 mM SPP buffer pH 7.4 and subjected to electrophoretic separation for 20 h at 40 V and 400 mA with buffer circulation, before staining with 0.2% (*w*/*v*) Coomassie Brilliant Blue R250 for 1 h and extensive washing with destaining solution (10% (*v*/*v*) acetic acid, 40% (*v*/*v*) ethanol).

#### 2.2.2. Transmission Electron Microscopy (TEM)

Virus samples were diluted in SPP buffer pH 7.0 (virus concentration: 200 ng/µL) and adsorbed to briefly ethanol-treated (i.e., hydrophilized) Formvar^®^/carbon-coated copper grids [87] for 10 min, washed three times for 1 min each on a drop of ultrapure deionized water, negatively stained with four drops of an aqueous solution of 2% (*w*/*v*) uranyl acetate, or 1% (*w*/*v*) uranyl acetate (UAc) supplemented with 250 μg/mL bacitracin (three drops ≈ 5 s, one drop, 5 min). After removal of the stain solution by filter paper, the sample was dried and analyzed using a Tecnai G2 Sphera transmission electron microscope (FEI, Eindhoven, The Netherlands) at 120 kV with a 16-megapixel camera (TemCam-F416R, TVIPS, Gauting, Germany), or with a ZEISS EM 10A at 60 kV with a 1-megapixel camera (TRS Slowscan, Albert Tröndle Restlichtverstärkersysteme, Moorenweis, Germany). The images were evaluated with Image J software version 1.46r [88]

#### 2.2.3. Gold Labeling of TVCV_PA_ via Antibody Capture

TVCV_PA_ (12 µg/µL in 10 mM SPP pH 7.2) or, as a control, TVCV_WT_ particles of similar concentration, were diluted directly to 0.3 µg/µL (f.c.) in 15 nm gold nanoparticle-conjugated rabbit anti-goat IgG antibodies (AURION Immuno Gold Reagents & Accessories, Wageningen, The Netherlands) and incubated for 1 h at RT. It should be noted that an IgG–gold conjugate preparation was applied that had been stored at 4 °C for years before use (see Section 4). Samples (10 µL) were then placed on the hydrophilized carbon layer of coated grids for 5 min. Following sample removal by filter paper, grids were washed six times on 10 µL drops of ultrapure deionized water, stained for TEM negatively by UAc as described above, and air-dried.

#### 2.2.4. Antibody Immobilization on TVCV_PA_ in Solution and Pull-Down Assay

An amount of 150 µg TVCV_PA_ (with 4.4 nmol PA domains), or 150 µg TVCV_WT_ as the negative control, was agitated with 300 µg rabbit anti-penicillinase antibody (IgG [≈2 nmol], GTX40959, GeneTex, Biozol Diagnostica Vertrieb GmbH, Eching, Germany) in 150 µL SPP 10 mM pH 7.0 for 2 h at 30 °C and sedimented by UC (4 °C, 1.5 h, 34,000 rpm ≈134,000× *g* [RCF_max_], 45 Ti rotor, Beckman Coulter Optima L90K). As a further control, 300 µg of the IgG alone were processed in an equal volume SPP in parallel. The supernatants (SN) were collected, and pellets (P) were resuspended in 150 µL 10 mM SPP pH 7.0. An amount of 1.5 µL of both fractions was analyzed via 12% SDS-PAGE under non-reducing conditions, i.e., without sample heating prior to loading.

### 2.3. Antibody-Mediated Display of Functional Molecules on TVCV_PA_ on Solid Supports

#### 2.3.1. IgG–Enzyme Conjugate-Coupling and Enzyme Activities on Dot-Blots

The activities of antibody-conjugated enzymes after coupling to viruses bound on nitrocellulose membranes were tested by spotting 0.5 µL of 1 µg/µL virus-containing solutions (TMV, TVCV_WT_, TVCV_PA_) onto the membrane, followed by air-drying for 5 min. After blocking with 1% BSA (*w*/*v*) in TBS for 1 h and three washing steps with TBS-T, membranes were incubated with either anti-goat antibodies from rabbit conjugated with AP- or HRP-conjugated rabbit antibodies (see above) diluted 1:500 in 2% BSA (*w*/*v*) in PBS at RT for 1 h. After washing (6 × 5 min in TBS-T), AP-activity was visualized via NBT/BCIP, and HRP-activity via 4-chloro-1-naphthol (4CN) staining solution.

#### 2.3.2. Microtiter Plate Coating with TVCV Variants and IgG or IgG Conjugate Binding

An amount of 5 µg TVCV_PA_ (≈150 pmol PA domains) or TVCV_WT_ control particles were diluted in 100 μL binding buffer (10 mM SPP pH 7.8, 137 mM NaCl), loaded into the wells of appropriate F-bottom, 96-well polystyrene high-binding microtiter plates, and incubated for 16 h at 4 °C. In the case of fluorescence assays, black plates were applied (FluoTrac MICROLON^®^, black, No.655077, Greiner Bio-One, Frickenhausen, Germany), for colorimetric detection clear ones (MICROLON^®^, clear, No.655061, Greiner Bio-One, Frickenhausen, Germany). After three washes at RT, each for 5 min with phosphate-buffered saline pH 7.4 (PBS) [80], wells were blocked with 200 μL 2% bovine serum albumin (BSA, *w*/*v*) in PBS (1 h at RT) and washed as before. An amount of 100 µL rabbit IgG antibodies or antibody conjugates of choice (see following paragraphs) were incubated in the wells for 1–2 h at RT as indicated below, and excess or unbound IgGs were removed in three consecutive washing steps with PBS before subsequent detection or antigen coupling procedures.

#### 2.3.3. Detection of Bound Antibody Conjugates with Fluorescent F(ab′)_2_-Fragments

Antibody coupling to the PA domains of TVCV_PA_ was verified for four rabbit IgG–enzyme conjugates by fluorescently labeled secondary anti-rabbit antibody F(ab′)_2_-fragments. The rabbit IgG-enzyme conjugates were incubated for 1 h in the TVCV-coated wells of black microtiter plates: rabbit anti-goat IgG-AP [605-4562, Rockland Immunochemicals, Limerick, PA, USA]; rabbit anti-mouse IgG-AP ([A4312, Sigma-Aldrich, Steinheim, Germany]; rabbit anti-mouse IgG-HRP [A9044, Sigma-Aldrich, Steinheim, Germany], each diluted 1:100 in PBS during incubation, and rabbit anti-GOx IgG-HRP [ab31760, polyclonal, Abcam, Cambridge, UK] diluted 1:180 in PBS. TVCV_WT_-coated wells served as negative controls. Following the post-coupling washes, immobilized rabbit IgGs were detected by fluorescently labeled secondary anti-rabbit antibody F(ab′)_2_-fragments (goat anti-rabbit IgG F(ab′)_2_-Alexa Fluor^®^ 647, 111-606-003, Jackson ImmunoResearch Europe Ltd., Suffolk, UK; diluted 1:250 in PBS). After incubation in the wells at RT for 1.5–2 h and three consecutive washing steps with PBS all in the dark, fluorescence was measured spectrophotometrically (Spectrafluor Plus infinite M200 pro, TECAN; at λ_Ex_ = 630 nm; λ_Em_ = 667 nm).

### 2.4. Antibody-Mediated Immobilization of Enzymes or Control Protein on TVCV_PA_ on Microtiter Plate Surfaces and Activity/Functionality Measurement

TVCV_PA_ particle coatings in the wells of a clear microtiter plate were equipped with alkaline phosphatase (AP) or horseradish peroxidase (HRP), respectively, conjugated to the Fc-part of rabbit anti-IgG antibodies specified above. Enzyme–IgG conjugates were diluted 1:150 in PBS and incubated for 2 h at RT. Following three washing steps (as above), enzyme activities were determined after addition of 200 µL substrate solution containing either 1 mg/mL para-nitrophenyl phosphate (pNPP) in 10% (*v*/*v*) diethanolamine buffer pH 9.8 for AP, or 2.5 mM 2,2′-azino-bis(3-ethylbenzothiazoline-6-sulphonic acid) (ABTS), 50 mM NaOAc pH 5.0 and 0.5 mM H_2_O_2_ for HRP, by spectrophotometric read-out of the absorbance changes at λ_Abs_ = 405 nm over a period of 20 min directly after substrate addition.

A bi-enzyme cascade of GOx and HRP was established on TVCV_PA_ immobilized in plate wells via incubation with rabbit anti-GOx IgG-HRP conjugate (diluted 1:180 (≈30 pmol) in 100 µL PBS) at RT for 1 h, post-binding washes and reaction with 20 µg GOx (≈125 pmol) in 100 µL PBS for 2 h at RT. After washing, a 200 µL substrate mix with glucose (100 mM glucose, 5 mM ABTS, 50 mM NaOAc pH 5.0) was added per well, and enzyme activities were determined as described above. The ABTS* concentration c was calculated from the absorption A at λ_Abs_ = 405 nm using the Lambert-Beer law $c=\frac{A}{d\cdot \varepsilon_{405}}$ with d = 0.625 cm (filling level of the well) and $\varepsilon_{405}nm=\frac{36.8}{mM\cdot cm}$ (extinction coefficient of ABTS* according to the supplier). Product concentrations plotted vs. time and turnover rates of ABTS were determined from the slopes of the linear sections. If plates were applied repetitively, these were stored wet at 4 °C in between the tests. For control purposes, Emerald green fluorescent protein (emGFP) was captured and detected on the TVCV_PA_ surface as follows: 100 µL rabbit anti-GFP IgG (Sigma SAB4301138, Sigma-Aldrich/Merck KGaA, Darmstadt, Germany; diluted 1:40 in PBS, corresponding to ≈ 53 pmol IgGs) were agitated in TVCV_PA_-coated, BSA-blocked wells of black microtiter plates at RT for 1.5 h, excess IgGs removed via washing, and the wells incubated with 33 µg (≈1 nmol) emGFP (hexahistidine-tagged streptavidin fusion protein, kindly supplied by Klara Altintoprak; as described [72]), in 100 µL PBS for 1 h in the dark. After washing with PBS, emGFP fluorescence was detected spectrophotometrically (as above) at λ_Ex_ = 470 nm; λ_Em_ = 509 nm.

### 2.5. Sensor Chip Preparation and Immobilization of PAH (poly(allylamine hydrochloride), TVCV_PA_, TVCV_WT_, and TMV

To compare the deposition of TVCV_PA_, TVCV_WT_, and TMV on sensor-chip surfaces, for scanning electron microscopy (SEM) analysis, multi-layer structures of Al/p-Si/SiO_2_/Ta_2_O_5_ were prepared, coated with PAH, or used directly for virion adsorption [36]. This chip arrangement is applied commonly as an electrolyte-insulator-semiconductor capacitor (EISCAP) structure for electrochemical biosensors [89]. The preparation, cleaning, and immobilization of virus particles directly on the chips were described in previous studies [44,45]. The EISCAPs were divided into two groups. One group underwent the virus-particle immobilization procedure directly after cleaning, whereas the other one was modified by an additional weak polyelectrolyte layer of poly(allylamine hydrochloride) (PAH) on the Ta_2_O_5_-gate surface of the EISCAP [90,91], prior to the immobilization of TMV, TVCV or TVCV_PA_, respectively, following a recently developed layer-by-layer (LbL) protocol [41]. In brief, the PAH powder (~70 kDa, abcr GmbH, Karlsruhe, Germany) was dissolved in 100 mM NaCl to 50 µM (final concentration, f.c.), and the pH was adjusted to pH 5.4 by NaOH. An amount of 100 µL of this PAH solution was applied to the EISCAP chips. The method for immobilizing TMV [41] was utilized for the loading of TVCV and TVCV_PA_ particles. Briefly, the surfaces of EISCAP chips (treated with PAH or non-treated) were exposed to 50 µL of 0.1 mg/mL TMV_Cys-Bio_ [41], TVCV_WT_, or TVCV_PA_ solution (in 10 mM SPP buffer, pH 7.2) for one hour. Residual non-adsorbed virus particles were washed away by 10 mM phosphate-buffered saline (PBS, pH 7.4). The chip surfaces were dried by N_2_.

### 2.6. Scanning Electron Microscopy (SEM)

The SEM analysis of the virus-treated EISCAPS was performed by a Jeol JSM-7800F Schottky field-emission microscope (JEOL GmbH, Freising, Germany). Prior to microscopy, a ≈5 nm thin film of Pt/Pd alloy (80:20) was deposited on the chip surfaces. This film is responsible for providing surface conductivity and preventing the accumulation of electric charges (charging effect) during the SEM analysis.

### 2.7. Statistical Evaluation

Statistical analyses were performed using the non-parametric Mann–Whitney Rank Sum Test (SigmaStat version 3.5 (Systat Software, Inc., San José, CA, USA)). A *p* value of less than 0.05 was considered to be significant (* *p* < 0.05; ** *p* < 0.01; *** *p* < 0.001). The data are presented as boxplots (lines: median values, box boundaries: 25/75% quartiles, whiskers: 100% percentiles). Graphs and diagrams were prepared by use of Inkscape (Software Freedom Conservancy, Brooklyn, NY, USA) and Graph Pad Prism 4 (Graph Pad Software Inc., San Diego, CA, USA).

## 3. Results

### 3.1. TVCV_WT_ and TVCV_PA_ Production, Isolation and Characterization

Both TVCV with wt CP, and TVCV_PA_ with *S. aureus* protein A domains D and E fused to the surface-exposed C-terminus of every CP subunit, were produced in systemically infected *N. benthamiana* after agroinfiltration of infectious viral vector constructs [57,76], or after mechanical inoculation of their leaf homogenates or virus particles purified thereof (i.e., after one passage). Initial TVCV_WT_ delivery used an *A. tumefaciens* clone with pICH17501 encoding the full viral genome [57]. For TVCV_PA_ inoculation, two *A. tumefaciens* clones with deconstructed viral pro-vectors (pICH20701, pICH21767) were combined with a recombinase-expressing third clone (pICH14011) as described [57].

#### 3.1.1. Both TVCV_WT_ and TVCV_PA_ Accumulate Efficiently in *N. benthamiana*, but Only the Wildtype Virus Infects *N. tabacum*

Agroinfiltration of either inoculum typically resulted in infection of all *N. benthamiana* plants treated (see below). Newly expanding leaves exhibited deformations and occasionally mosaic symptoms from 8–9 dpi onwards in the case of TVCV_WT_, and about one week later (starting ≈ 16 dpi) with TVCV_PA_. Both virus variants also induced chlorosis and stunting, with the wt viral symptoms more pronounced, including additional necrotic patches and enhanced growth retardation (Figure 2A). After mechanical inoculation of homogenates from infected *N. benthamiana*, infection rates were 100% as well, with the onset of equal symptoms two to five days earlier in most cases. SDS PAGE of crude leaf homogenates confirmed high levels of viral CPs in agreement with their expected molecular weights. Interestingly, mechanical inoculation of *N. tabacum* ‘Samsun’ nn plants resulted in 100% infection by TVCV_WT_ with leaf mosaic patterns similar to those on *N. benthamiana*, while TVCV_PA_ was not transmitted in three independent experiments with 6–10 tobacco plants each. SDS-PA gels excluded asymptomatic systemic infections, indicating that TVCV_PA_ is not infectious for *N. tabacum*.

#### 3.1.2. A Standard Tobamovirus Purification Procedure via Stepwise Enrichment Needs Adaptation for TVCV_PA_ and Yields the Expected Shortened Particles

Virus purification initially followed a standard protocol for TMV [81], based on stepwise virion enrichment from plant homogenates via the removal of contaminants by filtration and butanol extraction, and subsequent two-fold particle precipitation by slow addition of the water-soluble polymer PEG-6000 (4% [*w*/*v*] f.c.) in the presence of at minimum 2% (*w*/*v*; f.c.) monovalent cations based on [82]. Deviating from the original protocol, all steps were carried out at 4 °C to minimize protein degradation. Virions in the second PEG precipitate were washed and collected by ultracentrifugation [83]. This yielded up to 3 mg TVCV_WT_ per g leaf material and colorless opalescent suspensions. Under the same conditions, however, TVCV_PA_ came out to precipitate incompletely during the first incubation with PEG, resulting in greenish final products. Therefore, the respective supernatant was re-extracted with butanol and subjected to the subsequent treatments (i.e., second PEG precipitation and ultracentrifugation) as initial standard protocol, in order to purify both TVCV variants essentially by the same procedure. This yielded up to ≈2.5 mg virus/g leaf material for TVCV_PA_ and resuspended opalescent preparations comparable to those of the wt virus. The structure of both types of virus particles was evaluated by electron microscopy (Figure 2B) and native agarose-gel electrophoresis (Figure 2C). The sizes of the CP or CP-PA domain-fusion proteins were verified by SDS-PAGE (Figure 2D). As a reference, samples of TMV were analyzed in parallel.

TEM after negative UAc staining visualized the nanotubular shape of the viruses and, in the case of TVCV_PA_, a seam of intensely stained PA domains surrounding the outer longitudinal surface (Figure 2B). The particles’ lengths and diameters were determined by image analysis. This revealed a median length of ≈220 nm and a diameter of ≈18 nm for TVCV_WT_ (n = 126 nanotubes analyzed), and ≈270 nm median length/≈24 nm diameter for TVCV_PA_ (n = 396), i.e., below the length expectations for wt and PA-domain-exposing viruses, respectively (Appendix A). Only a negligible portion of either virion type had the expected full particle length. The appearance and shortening of both TVCV types resembled the data obtained for similarly purified particles in previous work, where a correlation between reduced particle length and the homogenization method was found [57,58]. Therefore, the detrimental effects of serial mechanical treatments during virus purification were addressed later in this study by developing an alternative purification protocol (see below).

In accordance with these results, native agarose gel electrophoresis of whole particles (Figure 2C) resulted in a band of TVCV_WT_ tubes with an electrophoretic mobility slightly higher than that of TMV monomers in a control sample separated in parallel. Whereas TMV formed a typical ladder of proportionally retarded di-and oligomeric head-to-tail virion aggregates, only a small fraction of TVCV_WT_ particles was found at the gel position expected for dimers. TVCV_PA_ migrated considerably more slowly on account of its larger CP molecular weight, size, and increased particle diameter (see Section 1).

SDS-PAGE of the purified samples revealed major CP protein bands corresponding to the molecular weight of 17.47 kDa for TVCV_WT_ and 33.05 kDa for TVCV_PA_ (wt CP plus linker-installed PA domains of 15.58 kDa). An amount of 1 mM PMSF, as well as 2 mM EDTA, were added to the TVCV_PA_ preparations as suggested earlier [57,58], but could not fully prevent some TVCV_PA_ degradation during storage.

#### 3.1.3. Virus Isolation from Crude Precipitates by Selective Inverse PEG-Sucrose Solubility Gradients Reduces Effort and Improves Particle Integrity

Recent evidence has indicated that mechanical treatments upon plant disruption affect the integrity and particularly the length of TVCV particles significantly [58], and as the precipitation of TVCV_PA_ from plant homogenates by 4% PEG-6000 and up to about 2% monovalent cations according to standard methods for tobamoviruses was not efficient, we evaluated an alternative PEG-based purification strategy. This ensured complete and fast virion precipitation directly from plant crude extracts, avoiding substantial physical stress. Without prior filtration, stirring and any repeated resuspension steps, better-preserved particle structures were expected. Following PEG-mediated virion precipitation directly from raw homogenates, gentle and specific re-solvation upon centrifugation into inverse gradients lowering the PEG concentration slowly was shown to result in virus particle banding in those phases where PEG/salt conditions allow re-solvation according to their specific biophysical properties [85]. Due to the charge, mass, and shape deviations between TVCV_WT_ and TVCV_PA_, the initial PEG/salt concentrations inducing complete virion precipitation after addition to the clarified crude homogenate in 500 mM sodium potassium phosphate (SPP) buffer (pH 7.2) had to be determined experimentally for both viruses. This needed for TVCV_WT_ 5.5% f.c. (*w*/*v*) PEG-6000, and for TVCV_PA_ 8% f.c. (*w*/*v*) PEG-6000, respectively, both in the presence of 1% (*w*/*v*) f.c. NaCl, supplemented via two-fold stock solutions. Aggregated virions were concentrated by centrifugation for 15 min, and resuspended sediments were laid onto inverse PEG solubility gradients (with decreasing PEG-6000/increasing NaCl concentrations from top to bottom with several variations of gradient spreading). Following centrifugation at ≈25,000× *g* (RCF_MAX_) for 30 min, bands of plant components and re-solubilized virions formed. To determine contaminant and virus particle distribution throughout the complete gradient lengths, these were fractionated, and aliquots were analyzed via SDS-PAGE for virus CP accumulation. Fractions with the highest CP contents and purities were pooled, TVCV_[WT/PA]_ particles were transferred into a PEG-free buffer via centrifugal ultrafiltration, and products were analyzed via TEM.

TVCV_WT_ accumulated to the highest concentrations at 1.2–1.5% [*w*/*v*] PEG-6000, TVCV_PA_ at 6.2–6.5%, reflecting the considerably different virion properties. Accordingly adapted 9 mL gradients prepared from six stacked layers diffused into each other overnight (see Figure 3 and Section 2) typically yielded 4 mL of high-purity virions in buffer (from pooled fractions, concentrations of best fractions around 5 mg/mL), corresponding to in total 1–1.5 mg TVCV_PA_ or TVCV_WT_, respectively, per g fresh *N. benthamiana* leaf tissue. Notably, length distributions of these particles were in full agreement with the expectations for the encapsidated RNA genomes: As determined by TEM image analysis, the by far largest classes of TVCV_WT_ rod lengths were 280–300 nm (in 10 nm intervals; n = 677 nanotubes analyzed), those of TVCV_PA_ 300–320 nm (n = 894). In addition, numerous shorter virus particles were present as known for TMV, representing mainly immature, incompletely encapsidated, and broken viruses, particles containing subgenomic and optionally also mutated viral, and in some cases host nucleic acids (see discussion). In comparison to length distributions determined recently only for TVCV_PA_, after variable homogenization techniques preceding PEG-based virion enrichment [58], the abundance of rods in a single well-defined class of complete length was substantially increased (Appendix A). Elongated rods (due to terminally adsorbed additional RNA-free CPs or head-to-tail attached viruses) were rarely observed.

### 3.2. Antibody Display on TVCV_PA_ as Plant Virus-Based Immunosorbent Nanoparticle (VIN)

A broad body of literature has shown that *S. aureus* protein A (PA) binds with high affinity to the constant regions (Fc) of many subclasses of immunoglobulin G (IgG) molecules from different species, with the highest binding efficiencies for human and rabbit IgGs [92,93,94,95]. TVCV_PA_ particles with two PA-derived immunoglobulin-binding domains exposed per CP have been exploited for the enrichment of human, humanized, and mouse IgGs from serum or plant origin in previous studies [57,58]. Here, we sought to apply TVCV_PA_ as VIN for rabbit IgG antibodies and antibody–enzyme conjugates in different configurations, in order to install enzymes and cooperating enzyme ensembles on the viral adapter rods. Goat IgGs were included in the study as well.

#### 3.2.1. Selective Binding of Rabbit IgGs to Purified TVCV_PA_

To validate the binding of purified TVCV_PA_ particles and rabbit IgG antibodies, pull-down tests with a molar ratio of PA-domains to IgG of 2.2:1 (4.4 nmol PA/2 nmol IgG) were carried out in 10 mM SPP pH 7.0, using TVCV_WT_ as a negative control for the specificity of antibody attachment. Following incubation of the reaction mixtures, sedimentation of virions and virion–antibody complexes by UC yielded pellets (P) and supernatants (S) analyzed by non-reducing SDS-PAGE (Figure 4A) and by native gel electrophoresis. These assays showed about 60% of the IgGs (≈1.2 nmol) bound by TVCV_PA_ in the pellets (Figure 4A red box), as determined by image analysis from the relative pixel intensities of the respective bands. This would correspond to one IgG on every fourth PA moiety or ≈550 antibodies per full-length TVCV_PA_ nanorod (≈2250 PA domains/317 nm).

Control TVCV_WT_ particles did not pull down any IgGs, and IgGs without binding partners did not precipitate under the conditions applied. Therefore, we conclude that the sedimentation of rabbit IgGs in the presence of TVCV_PA_ was due to specific binding to the surface-exposed PA domains. In addition, the decoration of TVCV_PA_ particles with antibodies resulted in large complexes preventing their migration into 0.8% agarose gels.

#### 3.2.2. Antibody Fc Conjugates with Enzymes or Gold Nanoparticles Bind to TVCV_PA_

TVCV_PA_ was adsorbed to microtiter plate well surfaces and then tested for its ability to capture different rabbit IgG–enzyme conjugates, with either alkaline phosphatase (AP) or horseradish peroxidase (HRP) coupled to their Fc–portions (rabbit anti-goat IgG-AP, anti-mouse IgG-AP, and anti-mouse IgG-HRP). Successful immobilization was detected by secondary F(ab′)_2_-fragments coupled to a fluorophore (goat anti-rabbit IgG F(ab′)_2_-Alexa Fluor^®^ 647), as non-human F(ab′)_2_ regions were reported to bind PA-domains with 50 to 100 times lower affinity than IgG Fc regions [96]. Accordingly, only in the presence of initially captured full IgG antibodies strong fluorescence was retained in the wells after extensive washing, whereas treatment with equal amounts of fluorescent F(ab′)_2_ fragments in the absence of IgGs did not stain the TVCV_PA_-coated wells to a considerable extent (Figure 4B shows relative fluorescence unit (RFU) values at λ_Em_ = 667 nm).

The capacity of TVCV_PA_ to bind antibody conjugates was investigated additionally by direct incubation with IgG–gold conjugates and TEM analysis. TVCV_WT_ and TVCV_PA_ particles were agitated with 15 nm gold nanoparticle-conjugated rabbit antibodies in solution for one hour and immobilized on carbon-coated TEM grid support films. Following washing and UAc staining, only TVCV_PA_ virions were found decorated with gold, demonstrating that despite the relatively large diameter of the gold beads on the IgGs’ Fc-portions, their binding to PA was possible, although with only single or few events detectable per virus nanorod (Figure 4C; see discussion below).

### 3.3. Display of Functional Enzymes on TVCV_PA_ Nanocarriers

#### Rabbit and Goat Antibody-Conjugated Enzymes Remain Active after Coupling to TVCV_PA_ Nanocarriers

Plant tissue-print blots on nitrocellulose membranes were initially performed to verify infection by the TVCV variants. Rabbit anti-TMV IgGs cross-reactive with TVCV served as primary, and AP-conjugated goat anti-rabbit IgGs as secondary antibodies (Figure 5A top). TVCV_PA_-containing stem or leaf prints produced stronger signals than those with TVCV_WT_, or one microgram purified TMV applied as a positive control (Figure 5A,B), which suggested additional direct binding of the enzyme–IgG conjugates to the PA domains displayed on TVCV_PA_. This was confirmed by tissue-print membranes incubated with the goat IgG-AP conjugate alone, yielding strong signals for TVCV_PA_ only (Figure 5A,B; each: bottom). The binding and catalytic activities of these goat IgG–conjugates and two further rabbit IgG–enzyme conjugates were validated on spots of purified TVCV_PA_, TVCV_WT,_ and TMV particles. Goat anti-rabbit IgG-AP (Figure 5A,B), rabbit anti-goat IgG-AP, and rabbit anti-mouse IgG-HRP all attached to TVCV_PA_ directly and specifically, with the enzymatic reactions detected via stain precipitates from BCIP/NBT substrate for AP, and from 4CN for HRP (Figure 5C).

Semi-quantitative tests were set up in microtiter plates, to assess antibody binding by TVCV_PA_ and corresponding enzyme activities. The two TVCV variants or TMV, respectively, were adsorbed to the surfaces of high-binding polystyrene plate wells, and—after blocking with BSA—incubated with rabbit anti-goat-IgG-AP or rabbit anti-mouse-IgG-HRP. Bound IgGs were detected via fluorescent F(ab′)_2_ fragments, and enzyme activities by chromogenic substrates (Figure 6). Tobamovirus-free cavities with or without BSA coating served as parallel controls.

IgG-conjugate binding, as revealed via goat anti-rabbit IgG F(ab′)_2_-Alexa Fluor^®^ 647-fragments, was specific for TVCV_PA_, with negligible fluorescence due to non-specific IgG or F(ab′)_2_-fragment adsorption to plate or PA-free virion surfaces (Figure 6A).

The concentrating effect of TVCV_PA_ was confirmed by high enzyme activities in the respective wells. The turnover rates of pNPP as substrate for AP, or ABTS and H_2_O_2_ as substrates for HRP, showed preserved activities of the Fc-coupled enzymes, even after binding to protein A (Figure 6B). Several control layouts were applied to discriminate selective bioaffinity binding of IgGs to PA from potential non-specific adsorption to the wells, all attesting to the specific adapter functionality and high binding capacity of TVCV_PA_ (Figure 6 and its legend).

### 3.4. A Two Enzyme-Cascade of GOx and HRP Installed on TVCV_PA_ via the Capture of a Single Antibody-Conjugate

After initial tests with rabbit anti-GFP IgGs had verified that TVCV_PA_-presented antibodies retained their capability for target capture (Appendix A), a two-enzyme cascade of GOx and HRP was established on the viral nanocarriers by way of a single IgG-conjugate type. Following TVCV_PA_ immobilization in microtiter plates, HRP-coupled rabbit anti-glucose oxidase antibodies (rabbit anti-GOx IgG-HRP) were bound to the CP-exposed PA domains via their IgG Fc regions, and equipped with GOx via their antigen-binding sites consecutively, with intermittent washing. Successfully combined cooperating active enzymes were expected to allow colorimetric glucose detection via a cascade reaction of GOx and HRP in the presence of a chromogenic co-substrate (see below).

#### 3.4.1. Anti-GOx Antibodies Conjugated with HRP Are Immobilized to Superior Amounts in Microtiter Plates via TVCV_PA_ Adapters

First experiments in this context analyzed the amounts of rabbit anti-GOx IgG-HRP in high-binding microtiter plate wells in the presence or absence of TVCV_PA_, and the influence of BSA for blocking non-specific IgG adsorption sites, in comparison to the capture of the rabbit IgGs applied in the above experiments (rabbit anti-mouse-IgG-HRP) (Figure 7). IgG immobilization was detected by fluorescent F(ab′)_2_-Alexa647. The experiments revealed that wells coated with TVCV_PA_ outcompeted all constellations devoid of viral adapters, even after blocking non-specific IgG binding sites with BSA prior to the wells’ exposure to rabbit IgGs.

The consecutive application of TVCV_PA_ and rabbit anti-GOx IgG-HRP without intermediate BSA deposition (1 in Figure 7A,B) resulted in the highest IgG load, due to both specific IgG binding to PA-domains and simultaneous non-specific adsorption to accessible well surface areas. BSA incubation after TVCV_PA_ immobilization (2) reduced the amount of IgG-HRP conjugate retained in the wells by about one-third. However, it surpassed the quantity of directly adsorbed anti-GOx IgG-HRP in the absence of prior BSA blocking (3), and almost no antibody was bound on BSA-treated but virion-free plate surfaces (4). This indicates that the affinity between the PA-domains exposed on TVCV_PA_, and the IgG Fc regions is much higher than that between microplate surface and IgGs and that BSA does not interfere considerably with this interaction.

The amounts of the rabbit IgG-HRP conjugate directed against GOx and captured selectively by TVCV_PA_ were in the same range as those of rabbit IgG-HRP directed against mouse IgGs (5). Again, the selective IgG capture by TVCV_PA_ after BSA blocking of non-specific binding sites immobilized higher amounts than attached in non-blocked ‘high binding’ plate wells (6).

#### 3.4.2. TVCV_PA_-Displayed HRP-Conjugated AntiGOx–IgGs Tether Both Enzymes into a Durable Cooperating System

Rabbit anti-GOx IgG–HRP conjugates installed on TVCV_PA_ in microtiter plates were applied for immunocapture of GOx and investigated for cooperative activity of the two enzymes in the presence of glucose and the chromogenic co-substrate ABTS. Reaction rates and reusability were compared to those of the equal antibody-HRP conjugate/GOx system directly immobilized in plate wells. While the amount of selectively TVCV_PA_-coupled, IgG-assembled enzymes did not exceed that of the non-specifically adsorbed IgG-bound bi-enzyme system considerably, the viral nanocarriers stabilized the enzymes’ shelf life over prolonged periods of time with repetitive uses, as detailed in the following and shown in Figure 8.

Colorimetric detection of the cooperative enzyme activities addressed the second reaction step: GOx-catalyzed glucose oxidation to D-glucono-1,5-lactone with oxygen as electron acceptor goes along with the formation of hydrogen peroxide. This is reduced by HRP with ABTS as an electron donor, converting it into its stable blueish radical cation ABTS* that allows spectrophotometric detection of the reaction kinetics. Following GOx immunocapture by HRP conjugates either installed on TVCV_PA_ in microtiter plates or adsorbed directly to the well surface (both with or without BSA blocking; see comparative series of treatments in Figure 8A) and washing, a mixture of glucose and ABTS was added to study the activities of the enzymatic cascade layouts and control combinations (Appendix A).

Glucose turnover was detected only in the presence of both GOx and HRP (Figure 8B), and was highest in the presence of TVCV_PA_. Without BSA blocking prior to GOx addition (layout 1 in Figure 8), turnover rates up to 10 µmol ABTS/min were reached. After blocking non-specific attachment sites for GOx (layout 2), rates of ≈7 µmol ABTS/min were gained. These were, however, similar to the turnover rates in the absence of TVCV_PA_ without or with BSA-blocked well surfaces before GOx attachment (≈5–5.5 µmol ABTS/min in 3 and 4). Taken together, these findings indicate that a combination of TVCV_PA_ surface enhancement with its specific antibody conjugate capture, accompanied by additional non-specific adsorption (Figure 8A(1),B(1)), led to the highest reaction rates. Mainly selective conjugate binding to the TVCV_PA_-coated, otherwise blocked well surface resulted in reaction rates comparable to those in TVCV_PA_-free wells.

Control layouts further demonstrated that the GOx/ABTS conversion depended on the high-affinity attachment of both enzymes, as GOx adsorption to wells loaded with GOx-non-targeting IgG–HRP conjugates (+BSA) ended up only in a background below 1 µmol ABTS conversion/min (Appendix A, layouts 5 and 7). It was slightly higher in the absence of BSA due to non-specific GOx adsorption close to the IgG–HRP conjugates, but far below that of selectively bound GOx (layout 6). The basic functionality of the enzyme or enzyme conjugate preparations combined for the cascade reaction was demonstrated in mixtures of both in solution (Appendix A).

The coupling of enzymes to technical supports may stabilize their structure and activity, may allow applications under unfavorable environmental conditions, and facilitate separation from the reaction media. Many attempts of enzyme immobilization, however, end up in strongly compromised activities or complete loss of function, which often results from steric constraints or blockage of the active sites, or insufficient flexibility if the biomolecules are installed on solid surfaces [97,98,99,100]. Direct adsorption to microtiter plate wells thus bears a high risk of adverse effects on the enzymes’ functions and also does not allow a spatially well-defined arrangement of interacting partner molecules. In recent years, nanoscale viral carrier scaffolds have emerged as promising multivalent soft-matter supports that do not only enhance the surface available for enzyme coupling in predictable inter-molecular distances but also have exerted beneficial effects on the long-term stability and in some cases even activity of enzymes displayed [16,49,50,101,102,103,104].

Hence, the performance of the two-enzyme cascade established on TVCV_PA_ adapter rods in microtiter plate wells was investigated for stability and reusability after repetitive use over 25 days, with intermittent storage at 4 °C (Figure 8C). The activities retained with TVCV_PA_ adapters (layouts 1, 2) were compared to those of the same enzyme systems combined on antibodies directly attached to the well surface (layouts 3, 4 in Figure 8). All samples showed reduced turnover rates already after the first use. Without viral nanocarriers, these were below the detection level after 10 days of serial use. In the presence of TVCV_PA_, however, an activity of 25% was preserved even after 20 days. With regard to reasonable glucose sensitivities in relevant applications, e.g., in biomedicine or food quality control, this allowed a five-fold longer use of the microplate sensors.

### 3.5. PAH-Promoted Immobilization of TVCV_PA_ Layers at High Surface Densities: Towards Applications in Biosensor Devices

Among analyte detection procedures, biosensing layouts are most advantageous if they do not only exploit the high selectivity, sensitivity, and small size of bioreceptors but also utilize a compact setup and label-free read-out to allow real-time monitoring and/or application at the point of need. All these options are realized if conversion reactions by, e.g., enzymes immobilized on chip surfaces, are recorded in small or miniaturized devices, via detecting electrochemical changes arising from the generation or consumption of electrons or protons. Respective sensor chips equipped with TMV adapter coatings as enzyme-displaying nanomaterial have proven their beneficial properties repeatedly [41,42,44,48,49]. Hence, we sought to evaluate if TVCV_PA_ immobilization on such chips can follow the protocols established for TMV before. Virion adsorption was compared to that of TVCV_WT_ with reference to TMV, as all these particles differ in their surface properties. Tantalum pentoxide (Ta_2_O_5_) surfaces of EISCAP sensor chips, without or with a positively charged PAH polyelectrolyte film [41] were included in the tests.

SEM analysis of the virus particles adsorbed on the EISCAP surfaces showed a significant difference between the two chip groups: treated or non-treated with a PAH layer (Figure 9). For TMV on the bare Ta_2_O_5_ surface (without PAH), a homogeneous distribution with large vacant spaces in between was observed, which was the same for TVCV_PA_. In both cases, the particles lay randomly, with lengths between ≈30 nm to 1 µm or longer detectable, which indicates an occasional head-to-tail attachment of two or few virions also for TVCV_PA_. For TVCV_WT_, almost no particles were visible in the absence of PAH (Figure 9, left part). The analysis revealed that the binding of TMV, TVCV_WT,_ and TVCV_PA_ on Ta_2_O_5_ surfaces might be limited by, e.g., electrostatic forces (repulsion). Oppositely, on PAH-treated EISCAPs, a denser to full coverage of the virus particles was observed with only very few, small vacant areas. In some regions, virions also appeared stacked on top of each other. In conclusion, binding between all types of virus particles and the sensor chip surface was enhanced by a PAH interlayer, confirming and extending our previous findings for TMV particles [41]. TVCV_PA_ particles even formed more homogeneously distributed layers after LbL adsorption on the PAH-treated EISCAP surfaces than their wildtype or TMV counterparts (Figure 9, right), which is a promising point of departure for future studies with label-free biosensors.

## 4. Discussion

As the subgroup 3 tobamovirus TVCV has allowed the genetic fusion of *S. aureus* PA domains D and E to all of its CPs, nanotubular TVCV_PA_ virions ensheathed with more than 2200 IgG-binding protein fragments are accessible from plants [57] and have been exploited here for the antibody-assisted display of multiple enzymes or bi-enzyme systems. TVCV_PA_ and TVCV_WT_ control particles were obtained from systemically infected *N. benthamiana* following agroinoculation, as described [57,76]. Interestingly, the mechanical passage of TVCV_WT_ to tobacco (*N. tabacum* ‘Samsun’ nn) was possible as well but did not succeed with TVCV_PA_. *N. benthamiana* is a highly permissive laboratory host and therefore applied, e.g., for the efficient molecular farming of protein compounds, and for investigating virus infections [105]. It lacks functional RNA-dependent RNA polymerase 1 (RDR1) [106], which is otherwise involved in RNA-mediated gene silencing. Whether or not the capacity of TVCV_PA_ to invade this species but not tobacco, different from its parental virus, is correlated with the RDR1 mutation, has not been analyzed here.

Depending on the isolation protocol, one to three mg virus could be purified per g of fresh leaf tissue, similar to the yield of TMV [84]. For TVCV_PA_, however, a standard method for stepwise tobamovirus enrichment from solvent-extracted plant homogenates via two-fold precipitation with 4% PEG-6000 (*w*/*v* f.c.) based on [57,81,82], and post-purification by UC detailed in [83], did not result in clear and rich preparations as for TVCV_WT_. This reflects the deviant virion surface and, most likely, also the lower aspect ratio of TVCV_PA_, since depletion-induced particle precipitation by PEG coacervates is charge-, size- and shape-dependent. Highly anisotropic particles and such of small size (i.e., low Stokes radius) undergo phase separation and thus precipitate at lower PEG concentrations than larger and/or spherical ones [85,107,108,109,110], in the presence of 0.2 to 0.5 M NaCl (i.e., 1.2–2.9% *w*/*v*) in the underlying studies. Higher salt concentrations enhance precipitation less efficiently than elevated PEG concentrations. In our work, dual precipitation with 4% PEG and 2.0–2.7% monovalent salt ions (from buffers and added NaCl) yielded colorless preparations of both TVCV_WT_ and TMV, but for TVCV_PA_ dark-green pellets with considerable virion portions remaining in the first supernatant. After re-extraction with solvent, this fraction allowed PEG-induced TVCV_PA_ precipitation into clear pellets, and proceeding with the protocol applied also to TVCV_WT_. In accordance with previous data [57,58], electron microscopy revealed virion lengths below the expectations for the resulting TVCV_WT_ and TVCV_PA_ preparations. The latter particles were shortened to a lesser extent, which might be due to a stabilizing effect of the PA domain shell. Diameters were in agreement with the predictions, with the seam of PA moieties well-discernible by TEM.

As our long-term goals include routine uses of the TVCV_PA_, and the assembly of well-defined artificial enzyme complexes applicable, e.g., in single-particle studies [43], we intended to speed up its purification and improve particle integrity by avoiding a series of consecutive mechanical treatments. The selective re-solubilization of virions from PEG precipitates produced in only briefly clarified plant homogenates, as tested for common plant viruses initially more than 40 years ago [85,111], appeared to us a promising alternative strategy. Virion crude precipitates are allowed to re-dissolve under conditions meeting their biophysical properties, upon centrifugation into ‘inverse gradients’ of high to low PEG concentrations from top to bottom, which are stabilized by complementary sucrose content. Re-solvated high-purity virions form light-scattering zonal bands (Figure 3) if appropriate PEG concentrations are applied for precipitation and in the gradients. Our optimized protocols verified substantially different characteristics for the two TVCV types: While TVCV_WT_ solubilized between 1 and 1.5% [*w*/*v*] PEG, TVCV_PA_ needed above 6%. After gentle PEG removal, virions in both these preparations indeed exhibited length distributions in agreement with the expectations for largely intact particles [55,57]. A single, most abundant length class correlated with the nanotube length encapsidating the complete wildtype or engineered RNA genome (Appendix A). This is a considerable improvement also over recent advancements of the stepwise PEG-based enrichment procedure via gentler homogenization [58]. For both TVCV variants, shorter particles accompanying the fraction of complete virions reflect unavoidable breakage products, but also not (yet) fully encapsidated RNAs, and particles with subgenomic, mutant, or host RNAs, as typical of tobamoviruses in general [112]. The variety of non-infectious tobamovirus-like tubes with lengths below 300 nm generated upon the replication of full-length virions in plants has been investigated systematically for TMV already from the early stages of electron microscopy in the late 1930s onwards e.g., [113], and references therein. Apart from more prominent particle fractions containing the natural subgenomic viral RNAs [114], nucleoprotein rods with mutated and truncated viral RNAs may co-exist with the intact ones, due to trans-complementation of lacking RNA functions by the parental helper virus as described in [115,116,117], and many further studies. Finally, a portion of the TMV-like tubes incorporates host transcripts without viral OAs as by-products of the viral life cycle; for exemplary original and referencing work, see [31,118,119,120,121]. These previous findings are in line with recent data from a replicating in planta transcription-/TMV CP-co-expression system that confirmed the importance of the TMV OAs for efficient and uniform packaging of viral RNA by CPs via nucleation at a single site but demonstrated the CP’s capacity for encapsidating OAs-free transcripts as well, although the proportion of nonviral RNAs incorporated in parallel was not analyzed [122]. Interestingly and different from TMV, only minor amounts of head-to-tail aggregates were observed for either TVCV type by TEM or in native agarose gels, although some were found on sensor chips. Collectively, the data indicate that the VIN may offer advantages in applications demanding colloidal, well-dispersed nanocarrier particles.

The load of the TVCV-exposed PA fragments with IgGs was assessed from the pull-down experiments to ≈550 antibodies per full-length virion (Figure 4A). This points to the tightest packaging of an IgG sheath around the virion: The gyration radius of rabbit IgGs has been determined to ≈6.7 nm by small-angle X-ray scattering (SAXS) in solution [123], thus covering 140 nm^2^ under conditions of full rotability. A 317 × 24 nm TVCV_PA_ cylinder (23,900 nm^2^ curved surface area) could thus accommodate up to 170 freely rotatable IgGs so that attachment of a larger amount, as suggested here, necessarily restricts IgG dynamics and conformational changing, of which certain motions are relevant for antigen capture [124].

However, despite this presumed spatial confinement, the densely immobilized IgGs on their VIN carriers were well-suited for selective coupling of functional biomolecules and even antibody-tethered enzyme systems with single IgGs anchoring two enzyme types simultaneously. TVCV_PA_ coatings of high-binding microtiter plate surfaces increased the amounts of immobilized IgGs and enzymes considerably over those in VIN-free wells (Figure 6 and Figure 7). Notably, this was achieved via antibodies with the enzymes chemically coupled to their Fc region, i.e., close to the interaction sites with the PA domains. In initial proof-of-concept tests using IgG conjugates with gold nanoparticles of 15 nm diameter, only low decoration densities were obtained (Figure 4). However, in addition to putative steric constraints, the presence of non-conjugated IgGs in the preparation was not excluded and could thus have contributed to the inefficient labeling (see Section 2). With AP or HRP conjugates on goat or rabbit IgGs, though, high enzymatic activities could be immobilized on TVCV_PA_ (Figure 6 and Figure 7). Following additional immunocapture of GOx on the IgGs coupled to HRP, the cooperative substrate turnover actually surpassed that of the IgG-assembled bi-enzyme combination in BSA-free wells, and was in the same order of magnitude in wells blocked with BSA (Figure 8). This suggests that the coupling method did not compromise enzyme reactivity to a major extent. Most importantly, however, repetitive testing of the glucose detection cascades on TVCV_PA_ revealed higher remaining activity from the third day on, in comparison to those adsorbed directly in microtiter plates (Figure 8). TVCV_PA_ imparted a four-fold longer reusability until glucose sensitivity fell below 25% of the initial values, independent of the presence of BSA. Evidently, TVCV_PA_ adapter particles possess biomolecule-stabilizing properties similar to TMV in previous work [46,49,50], despite their deviant surface composition. This renders them promising candidates for use as adapter coatings on detector chips in biosensor setups applied for electrochemical, label-free real-time analytics, as established with TMV [41,42,44,48,49]. Recent methods for the electrostatic LbL adsorption of TMV on polyelectrolyte-modified Ta_2_O_5_ chip surfaces could be adopted one-to-one and yield even, nearly closed TVCV_PA_ layers. High surface densities of anchoring sites for recognition elements on EISCAPs are a key prerequisite for sensitive analyte detection [41,42,44]. Hence, this study has provided all methods and building blocks for on-chip manufacture of versatile TVCV_PA_-IgG-supported bioreceptor layers. They are expected to have one-of-a-kind benefits over most currently applied immobilization platforms in biosensor devices: multiple, repetitive analyte capture sites exposed at nanometric distances, with the coupled receptors ‘programmable’, i.e., defined by the IgGs’ affinities. These may bind targets either directly for, e.g., label-free field-effect [89,125,126] and impedance-based [38] detection, or install sensor enzymes generating electrochemically detectable components [42,44,48,49]. Upon accumulation in close vicinity to the field-effect sensor chip, these alter its surface potential progressively and thus lead to signal amplification. Tobamovirus-mediated bioreceptor stabilization on sensor chips may even exceed that in microtiter plates as observed for TMV-displayed GOx [49]. Accordingly, the prepared sensors can not only be stored for extended periods before use but they also allow in-house calibration prior to on-site application, which is of great advantage, e.g., for environmental monitoring and other uses in remote sensing. The TVCV_PA_-assisted configuration now awaits evaluation of its performance and applicability for different analytes.

Plant viral scaffolds are developed into biotechnically useful enzyme nanocarriers in several labs worldwide. One branch in this research area is the use of virus capsids and VLPs as cages confining enzymes inside, with exciting options for tailoring biocatalytic pathways and reaction parameters [127,128]. The second main branch employs viral backbones as immobilization support. Apart from TMV-based approaches presented initially in 2015 [50] and summarized in the introduction, pioneering earlier studies and recent research have used a number of flexuous and icosahedron-based virions as high surface-area scaffolds for various enzymes [16,25,101,102,103,104,129]. If enzymes are installed on CPs, the type of linkage may be a key to functionality, durability, and both effort and versatility of loading: Directed attachment and sufficient free space below and around the enzyme can assure accessibility of the active sites. Strong or covalent coupling may stabilize the activity [97], whereas low-affinity binding can help to exchange enzymes and thus regenerate or modify the system, but bears a risk of leach-out [130]. Finally, control over the surface density and spatial distribution of individual or multiple enzyme types are desired, to tailor the collective overall activity and to design artificial multi-enzyme complexes or multitasking configurations [131]. Against this background, immunoaffinity capture is attractive: It confers selectivity so that the functionality of choice can be installed site-specifically from raw preparations or molecule mixtures, and the IgGs act as 15–25 nm spacers with sufficiently tight target linkage.

Naturally produced high-affinity IgGs typically bind antigens with equilibrium affinity constants K*_a_* of 10^7^ M^−1^ up to 10^10^ M^−1^ as detailed in [132], and references therein—binding strengths that hold the target in place under a wide range of conditions, but allow induced release, e.g., by pH of 1.0–3.0, or chaotropic compounds [133], and IgG regeneration. If the IgG anchors are installed themselves via bacterial proteins A, G, L, IgG-binding domains thereof, or other respective ligands [134], however, the potentially lower strength of this junction has to be considered: For the PA two-domain (D/E) construct on TVCV_PA_, a K*_a_* above 10^7^ M^−1^ was determined for its binding to IgG Fc, and a K*_a_* of ≈3 × 10^5^ M^−1^ for Fab [96]. For complete PA with its five IgG-binding domains, K*_a_* values between 10^8^ M^−1^ and 10^9^ M^−1^ were described [135,136]. In our work, the PA domain D/E construct on TVCV_PA_ has achieved reliable IgG retention in all assays, as well as in crude plant homogenates tested before [57,58]. As the K*_a_* of PA domains D/E for IgGs is lower than that of many IgGs for their targets, refurbishment of the TVCV_PA_ nanocarriers with fresh sensor enzymes will, in most cases, need replacement of the complete IgG–enzyme complex. A pH drop down to pH 3 will likely suffice to this end, which is tolerated by tobamovirus particles [137,138], but remains to be tested systematically.

The total effort for the manufacture of enzyme-displaying particles is relatively low for TVCV_PA_, which also offers further advantages in comparison to other nanostructures and connection strategies for enzymes to generate surface-enhanced materials and improve handling. Plant viral backbones are generally sustainably produced and robust but biodegradable, and possess large numbers and regular arrangements of genetically and/or chemically addressable sites at nanometric spacing. Convincing activities and manipulation properties were found for most plant virus-based biocatalytic nanomaterials described above. Among those, PA- or PA domain-decorated viral scaffolds with IgGs mediating enzyme capture are convenient, because their preparation does not involve any costly chemical linkage (except for the special case of IgG–enzyme conjugates), as it has been necessary in several previous studies [16,41,43,44,46,48,49,50,139]. Moreover, the concept also does not involve genetically engineered, heterologously expressed, and purified enzymes exposing ligation-competent or affinity tags complementary to virus-displayed partner moieties [25,51,140]. These may be of great advantage for the irreversible display of routinely used enzymes but lack flexibility and ease if variable and multiple distinct enzyme ligands are needed, as typically is the case in biosensor applications.

Only a small number of studies have, though, exploited the affinity of PA domains for immobilizing antibodies on plant viral carriers, and the over-installation of an outer shell of thereby captured biofunctional molecules is—to our knowledge—an unprecedented concept. The size of PA domains often excludes their direct genetic fusion to the CPs of systemically infectious viruses so that delivery by agroinfiltration for local accumulation, and thus additional expenditure and equipment would be inevitable for in planta production. A way out occasionally followed is a heterologous expression of the recombinant CPs and assembly of virus-like particles (VLPs), which, though, yields less defined lengths than naturally RNA-scaffolded species and needs further preparative steps [129]. Notwithstanding, some IgG-binding VLPs have been generated successfully this way: The CP of potato virus M (PVM)—in the genus *Carlavirus,* family *Betaflexiviridae*—was equipped with a 78 aa fragment containing at least a single PA Z-domain, a genetically optimized IgG-binding polypeptide derived from the PA domain B [141,142]. The CP fusion yielded IgG-capturing flexuous VLPs [143]. Similarly, PA domain B-fused pepper vein banding virus (PVBV) CP assembled into *Potyvirus*-based filaments with high IgG binding capacity, intended for biomedical antibody delivery [144]. Whereas these CPs of elongated plant viruses were endowed with the PA domains at a surface-exposed terminus, another CP—of the spherical Sesbania mosaic virus (SeMV, genus *Sobemovirus*)—was engineered into a replacement mutant, with the PA domain B in place of a disordered CP region [145]. This recombinant CP assembled into unnatural icosahedral VLPs smaller (T = 1) than the RNA-containing parental virus, which was able to capture IgGs via the PA insertions.

The only other systemically plant-infectious, efficiently IgG-adsorbing PA domain fusion construct seems potato virus X (PVX, genus *Potexvirus*) presenting the PA domain B on its CPs [146]. These engineered, flexuous particles were shown to capture 300 to 500 IgG antibodies each. Since this hybrid was designed for biosensing the presence of IgGs, it has, though, not been applied for a subsequent immunoattachment of antigens. Such has been realized via a different approach with plant-produced potyviral filaments, which were actually equipped with enzyme shells immobilized by an IgG sheath—but in a mostly reverse configuration [101]: IgGs directed against zucchini yellow mosaic virus (ZYMV) served as docking sites for enzymes that were genetically fused to the PA derived, domain Z-deduced peptide Z33. The work achieved a ZYMV surface coverage of ≈87% with Z33-tagged, fully active 4-coumarate:CoA-ligase 2 (4CL2). The system was later expanded to a combination of both Z33-fused 4CL2 and stilbene synthase (STS), which catalyze consecutive steps in the synthesis of resveratrol from ρ-coumaric acid. Despite the efficient high-density display of the active enzyme cascade on the IgG-decorated potyviral potato virus A (PVA) backbone [104], low STS activity led to low resveratrol yield. This ‘reverse’ system is thus appealing for uses with extensively applied enzymes in optimized fusion with Z33, but demands genetic engineering and heterologous expression of each enzyme-Z33 hybrid protein before its functionality can be tested on the viral carrier—an effort not required with the TVCV_PA_ system.

Taken together, TVCV_PA_ particles have come into play as straightforward alternatives to other viral enzyme nanocarriers, due to their sustainable production in plants, protocols yielding well-defined preparations, and fast and versatile loading with IgGs capturing target enzymes of choice, including cost-efficient commercial preparations that will be immunoenriched on the tobamoviral nanorods. They exhibit a reliable capacity for the high-density display of antibodies on their PA domain-sheathed surface, as shown for different IgG types, and further beneficial properties that include the stabilization of biomolecules and the formation of even layers on sensor chip surfaces. This goes along with flexible options for multitasking and molecular cooperation, achievable via combinations of distinct targets on the multivalent particles.

## 5. Conclusions

PA domain-covered TVCV_PA_ particles were prepared from systemically infected *N. benthamiana* plants via optimized and novel methods, improving the VINs’ purity. The use of inverse PEG concentration gradients re-solubilizing the virions selectively from PEG precipitates of cleared raw homogenates shortened the process and increased particle integrity, which can have importance for single-particle applications and fundamental structural analyses. The PA domains allowed capturing hundreds of IgG molecules per nanorod, corresponding to a densely packed antibody sheath, which retained affinity to the antigen targets. This has been exploited for installing GFP and enzymes on the virus and offers versatile opportunities for immobilizing and combining functions of choice on a single bioscaffold by fast and efficient procedures, given that an appropriate IgG is available or can be raised. The use of the TVCV_PA_ particles as microtiter plate coatings enabled quantitative analyses, and revealed that they can serve as adapters for increasing well surfaces and mediating selective docking so that higher target loads can be achieved. This might become a practically relevant application of the VINs attractive for various microtiter plate-based assay formats. Notably, not only immunocaptured GOx remained active in the VIN-displayed configuration, but also enzymes conjugated to the Fc regions of the attached IgGs. This expands the TVCV_PA_-centered construction kit further, as it enables the use of IgGs as connector knots between two collaborating enzyme species and the virion backbone. Consequently, artificial multienzyme complexes with tethered GOx/HRP partners working in the cascade were generated by way of a single IgG type. This layout came out to stabilize the enzymatic activities over many repeated uses, paving the way to novel catalytic materials or modular biosensor layouts, with easy exchange and combination of distinct specific functionalities possible. This would be realized best in label-free sensor environments operated at the point of need, for which reason the applicability of TVCV_PA_ as receptor layers on a respective sensor was tested. In combination with a polyelectrolyte support film, the virions could be deposited as high-density and even surface coatings, laying the grounds for their integration into handheld sensor chambers.

In conclusion, this work has started the assembly of a powerful, multifunctional, and tailorable immunosorbent carrier platform, based on sustainably farmed biological nanoparticles. Future work might evaluate the TVCV_PA_ concept not only for biodetection tasks from food analysis up to environmental monitoring but also for in situ signal amplification and biomedical uses.

## Figures and Tables

**Figure 1 viruses-15-01951-f001:**
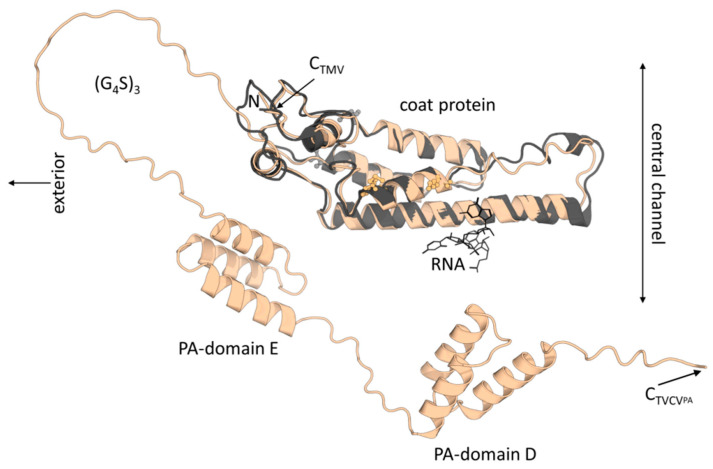
Comparison of TMV CP (grey) and TVCV_PA_ CP (apricot) structural organization (overlay scheme). The TMV CP tertiary structure (3J06) in assembled (metastable, calcium-free) virions [73] is overlaid with a TVCV_PA_ CP tertiary structure predicted by ColabFold [74]. Input for TVCV_PA_ CP: TVCV CP amino acid sequence [54] without N-terminal methionine, with C-terminal extension by a 15 aa linker [(G_4_S)_3_] preceding protein A domains E and D. This C-terminal fusion portion is shown in random orientation, neglecting its potential spatial arrangement on the virion surface. Lysines of wild-type CPs are shown as ball-and-stick representations in light grey (TMV) or light orange (TVCV). N: position of N-termini; C-termini as indicated.

**Figure 2 viruses-15-01951-f002:**
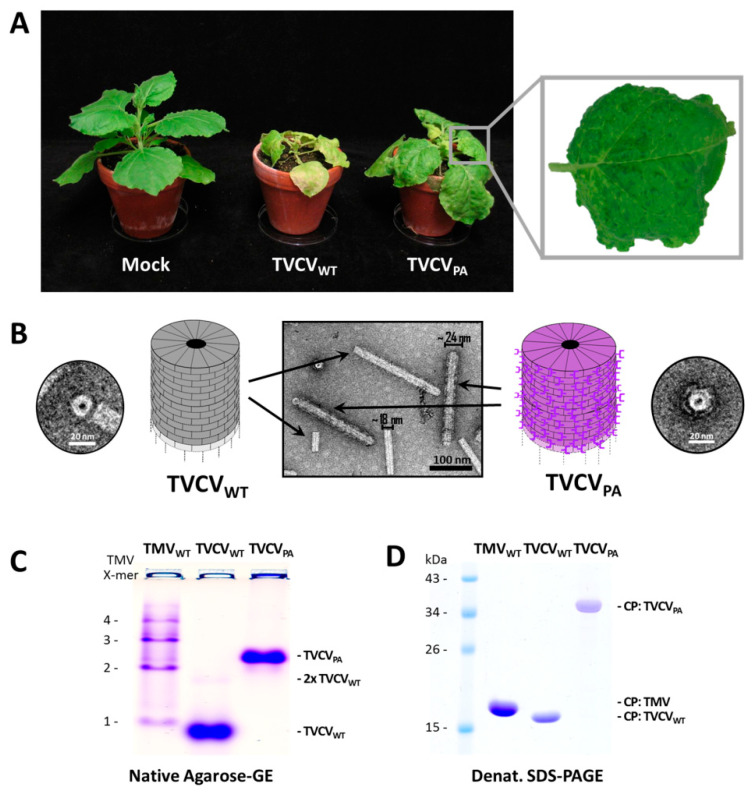
Purification of TVCV and TVCV_PA_ via stepwise enrichment from plant homogenates including two-fold PEG application and particle characterization. (**A**) Virus farming in plants; from left to right: *N. benthamiana* plants uninfected (mock), or infected with TVCV_WT_ or TVCV_PA_ after mechanical inoculation (13 dpi). Box: Leaf exhibiting a typical TVCV-associated mosaic. (**B**–**D**) Comparative analyses of TVCV_WT_ and TVCV_PA_ particles: (**B**) Center: TEM image of a mixture of TVCV_WT_ and TVCV_PA_ particles after negative UAc staining; left/right: schemes and higher magnifications of short upright virion fragments or CP nanoring (‘disk’) assemblies revealing the PA domain seam in the case of TVCV_PA_ (right). (**C**) Electrophoretically separated virus particles under native conditions in 0.8% agarose gels as indicated, stained with Coomassie-Blue R250 (refer to text for details). (**D**) 12% SDS-PA gel with CP bands of denatured virions and molecular weight marker bands as indicated (Coomassie-Blue stain).

**Figure 3 viruses-15-01951-f003:**
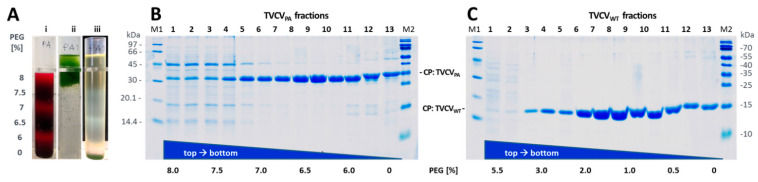
Isolation of TVCV_PA_ and TVCV particles via inverse PEG solubility gradients. (**A**) Gradient appearance (i) after ON diffusion, initial step fractions were alternatingly stained with acid fuchsine, PEG concentrations as indicated [% (*w*/*v*)]; (ii) after loading raw PEG precipitate; (iii) after centrifugation; ii/iii exemplified for TVCV_PA_. (**B**) TVCV_PA_ and (**C**) TVCV_WT_ fractions as indicated; SDS-PAGE (15% PA); Coomassie Blue-R250 stain.

**Figure 4 viruses-15-01951-f004:**
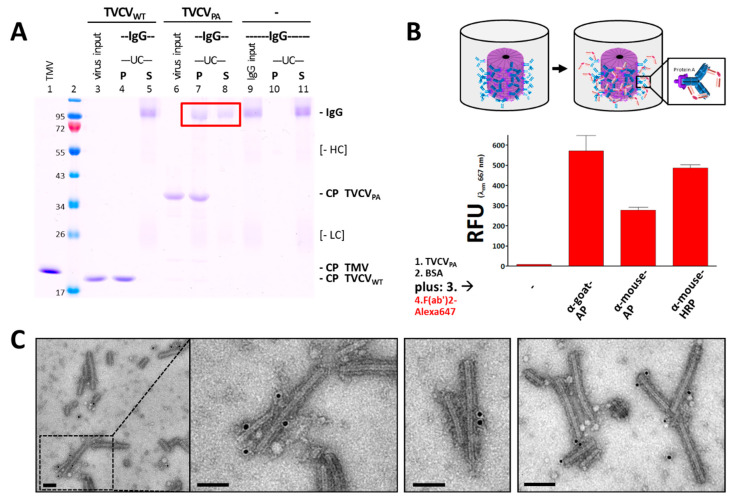
Antibody immobilization on PA-displaying TVCV particles. (**A**) Pull-down assays verify specific IgG attachments. Incubation of TVCV_WT_ or TVCV_PA_, respectively, with rabbit IgG antibodies in solution was followed by sedimentation of virus and virus–antibody complexes via UC. IgGs alone were treated in parallel. Resuspended pellets (P) and supernatants (S) were analyzed by non-reducing 12% SDS-PAGE (input lanes: loaded with input amounts of either virus (TVCV_WT_ or TVCV_PA_) or IgG without further treatment, as indicated). About 60% of the IgG input was bound by TVCV_PA_ and sedimented (red box). Positions of viral CPs, assembled IgGs, and minor amounts of released light chains (LC)/heavy chains (HC) are denoted. (**B**) Detection of rabbit IgG conjugate coupling to immobilized TVCV_PA_ in high-binding microtiter plates, via fluorescent secondary anti-rabbit IgG F(ab′)_2_-fragments. In TVCV_PA_-coated wells treated with BSA, different rabbit IgG antibody–enzyme conjugates [anti-(α)-goat or α-mouse IgG-AP or -HRP, as indicated) were applied. Bound IgGs were detected by secondary anti-IgG F(ab′)_2_–Alexa647 fragment conjugates. RFU = relative fluorescence unit. (**C**) Visualization of rabbit IgG display on TVCV_PA_ by TEM. Virions were decorated in solution with rabbit-anti-goat IgGs conjugated with 15 nm gold nanoparticles, deposited on grids, and stained by UAc. Scale bars: 100 nm (left images: overview and boxed area at higher magnification).

**Figure 5 viruses-15-01951-f005:**
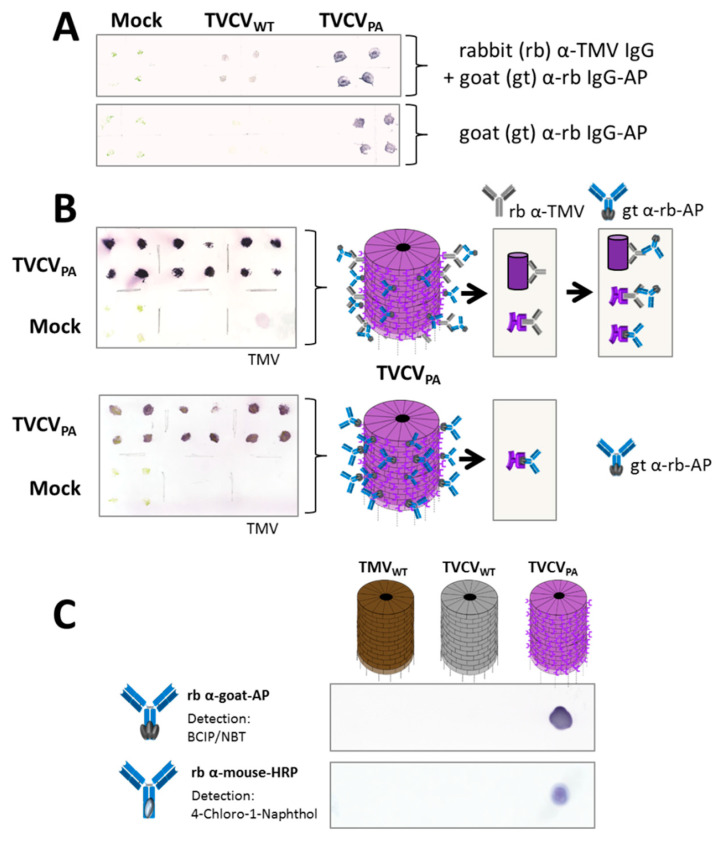
Plant tissue prints and virion spot-blots demonstrate selective binding of catalytically active goat and rabbit IgG–enzyme conjugates (AP and HRP) to membrane-immobilized TVCV_PA_. (**A**) Tissue prints probing infection of *N. benthamiana* with TVCV_WT_ or TVCV_PA_. Four randomly chosen leaves or leaf petioles of TVCV_PA_-inoculated plants were pressed shortly onto a nitrocellulose membrane, with tissues of mock-inoculated plants as a negative control. (**B**), as in (**A**), including spots of 1 µg isolated TMV as a positive control for the antigen homologous for the primary IgG. (**A**,**B**): Membranes were blocked with 1% BSA. *Upper blots:* Membranes incubated with rabbit anti-TMV IgGs, and secondary goat anti-rabbit IgG-AP conjugate; *bottom blots:* TVCV_PA_ after incubation with goat anti-rabbit IgG-AP only. (**C**) Spot blots of TMV, TVCV_WT,_ and TVCV_PA_ particles treated equally with rabbit anti-goat IgG-AP (*upper blots*) or rabbit anti-mouse IgG-HRP (*lower blots*) (see text for details). AP and HRP detection via stain precipitates generated from NBT/BCIP or 4CN, respectively.

**Figure 6 viruses-15-01951-f006:**
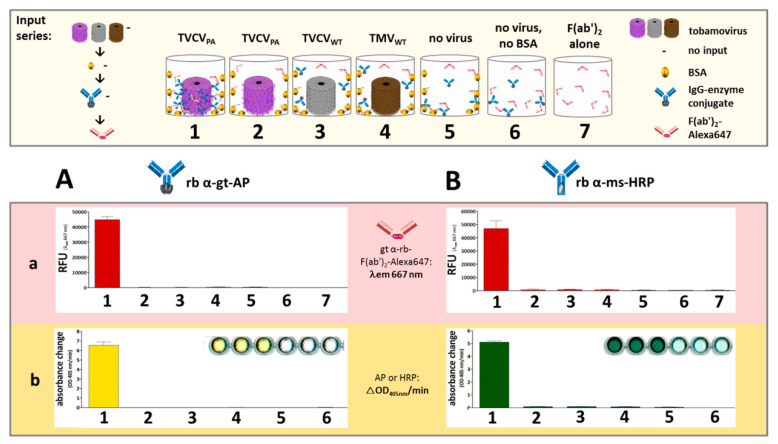
Coupling and enzymatic activities of antibody-conjugated AP or HRP immobilized with/without TVCV_PA_ in high-binding microtiter plates. Test series with (**A**) rabbit anti-goat-IgG-AP or (**B**) rabbit anti-mouse-IgG-HRP. Top: Series of components incubated in layouts 1–7: TVCV_PA_, TVCV_WT_, or TMV_WT_ particles (5 µg each), or none (virus-free control wells), as indicated; followed by washing, BSA, and IgG–enzyme conjugates (1:150), if depicted, and fluorescent goat anti-rabbit IgG F(ab′)_2_-Alexa Fluor^®^647. Legend at right and above wells. (**a**) Detection of immobilized IgG conjugates by fluorescence read-out at λ_Ex_ = 630 nm; λ_Em_ = 667 nm. (**b**) Spectrophotometric detection of antibody-conjugated enzyme activities via chromogenic substrates. AP-activity (**A**) was tested with 1 mg/mL pNPP, HRP-activity (**B**) with ABTS and 0.5 mM H_2_O_2_. Both reactions were monitored at λ_Abs_ = 405 nm over a period of 20 min. Insets in (**b**) each show three reactions with layout 1 and three reactions with layout 5.

**Figure 7 viruses-15-01951-f007:**
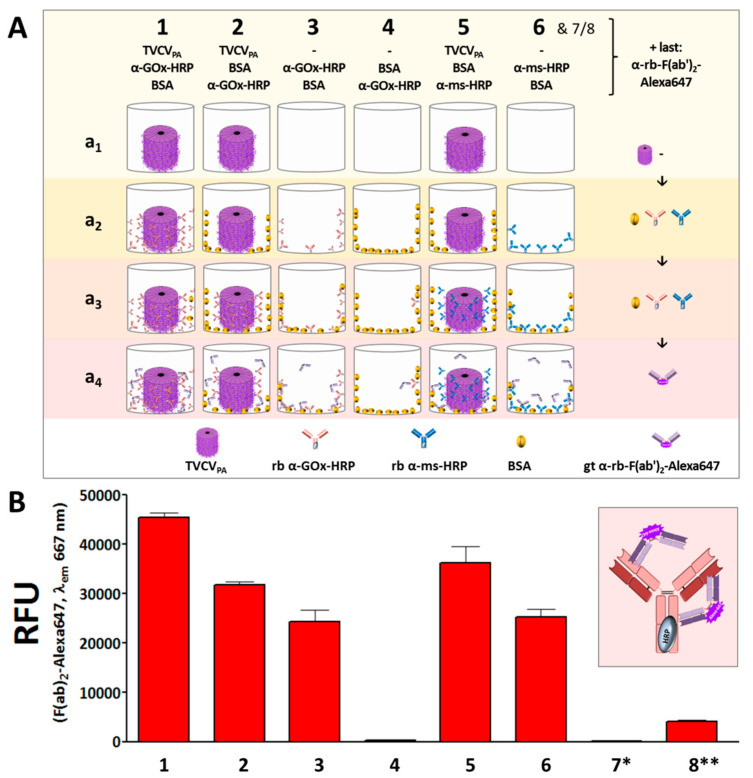
Coupling of antibody conjugates with HRP to TVCV_PA_ adapters in high-binding microtiter plate wells. (**A**) Scheme of comparative consecutive treatments (from a_1_ to a_4_) in six layouts (1–6) evaluating the amounts of selectively and non-specifically bound rabbit anti-GOx IgG-HRP (anti = α) or rabbit anti-mouse IgG-HRP, respectively, detected via fluorescent anti-rabbit-IgG F(ab′)_2_-fragments in wells with or without TVCV_PA_ coating, in the absence or presence of BSA, or for untreated supports, as indicated. */** Control layouts: 7/8: Controls with fluorescent F(ab′)_2_ fragments alone, applied with (7) or without (8) prior BSA treatment. In layouts 1,2 and 5, 5 µg virus particles were applied as well coating. (**B**) Immobilized IgG antibodies were detected by fluorescence-labeled anti-rabbit (α-rb) IgG F(ab′)_2_-Alexa647 fragments (inset); RFU: relative fluorescence units at λ_Ex_ = 630 nm; λ_Em_ = 667 nm. rb, ms, gt: abbreviations for rabbit, mouse, and goat, respectively.

**Figure 8 viruses-15-01951-f008:**
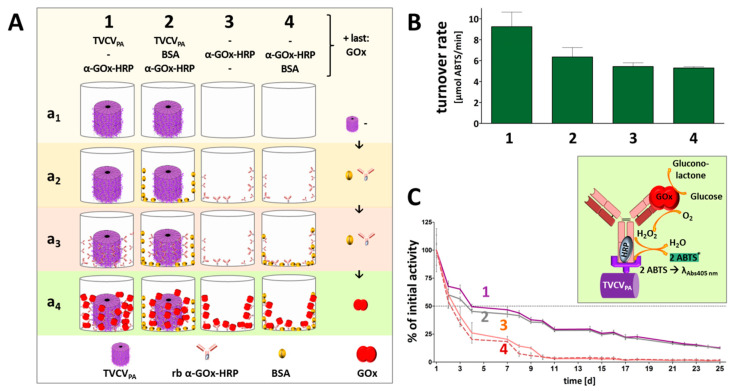
Antibody-mediated immobilization of the bi-enzyme cascade GOx/HRP in microtiter plates with or without TVCV_PA_ particles and BSA blocking, and corresponding enzyme activities. (**A**) Scheme of the main layouts compared for rabbit anti-GOx IgG-HRP-mediated immobilization of the cooperating enzymes GOx/HRP on well surfaces with or without immobilized TVCV_PA_ adapter particles. a_1_ to a_4_: Series of steps as indicated above, from top to bottom. In layouts 1 and 2, 5 µg TVCV_PA_ was applied for adsorption. (**B**) ABTS turnover rates achieved with the layouts in (**A**) with equal input of anti-GOx–IgG–HRP antibody conjugates and GOx molecules in all wells; below/right: reaction scheme of GOx and HRP installed on TVCV_PA_ via rabbit anti-GOx-IgG-HRP antibodies (see text for details), converting the chromogenic substrate ABTS into the colored product ABTS*. (**C**) Long-term reusability of the enzymes immobilized via the different layouts as shown in (**A**). Enzymatic activities were monitored over 25 days with multiple repeated uses. Initial activities of the immobilized enzymes as in (**A**,**B**) were set to 100% and the percentages of remaining activities after each use were calculated. For control layouts lacking GOx-specific IgGs, see Appendix A.

**Figure 9 viruses-15-01951-f009:**
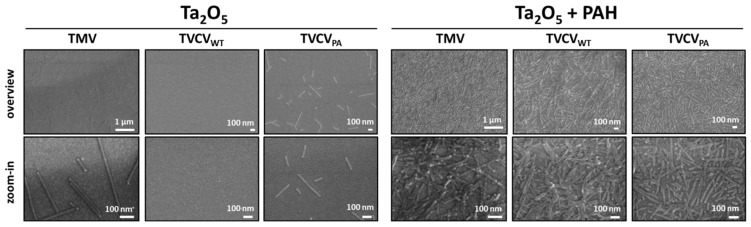
TMV, TVCV_WT,_ and TVCV_PA_ particle coverage on Ta_2_O_5_ sensor chip surfaces without and with PAH layer. SEM analysis following virion adsorption from 0.1 mg/mL stock solutions onto bare or PAH-coated Ta_2_O_5_ surfaces as indicated. Top rows: overview images taken at 20,000× (TMV) and 50,000× (TVCV_WT_, TVCV_PA_) magnifications, bottom: at 150,000× (TMV) and 100,000× (TVCV_WT_, TVCV_PA_) magnifications. For total magnifications, refer to scale bars. A 5 nm thin film of Pt/Pd alloy (80:20) was evaporated onto the specimens; imaging at 5.0 kV.

## Data Availability

Not applicable.

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
