# Peer review of "Facile Purification and Use of Tobamoviral Nanocarriers for Antibody-Mediated Display of a Two-Enzyme System"

_viruses, 2023, doi:10.3390/v15091951_

Round 1
Reviewer 1 Report
In the work, a particular chimeric viral of Tobamovirus is developed expressing the protein A of Staphylococcus aureus on its surface (TVCV-PA). They establish a new technique for the extraction of complete viral particles and use them as immunosorbents for IgG immunoglobulins. Based on immunoadsorption, the design of viral nanocarriers of enzymes and biofunctional molecules that allow the development and improvement of diagnostic methods (biodetection) is shown. The work proposes a new option in the development of viral nanocarriers and evaluates its applications.
Some specific observations are the following.
Line 133; Write the meaning of tem in this first appearance of the acronym.
Line 170; write the meaning of NTR
Line 387; although it appears later, writing the meaning of SEM in this his first writing
Line 430 and 431; write in italics the name of N. benthamiana and N. tabacum.
Line 446; in figure 2D, for CP:TVCV it is indicated that it is wt, however in TMV it is not indicated.
Line 519; since it appears here for the first time, indicate here the meaning of SPP

Author Response
We are happy about this very favourable assessment of the manuscript and for locating inconsistencies in the explanation of abbreviations and in formatting. We have corrected all of them in both text and figures, and hope to have also considered necessary corresponding changes elsewhere in the text/figures. Thanks a lot!
Specific responses:
Line 133; Write the meaning of tem in this first appearance of the acronym:
done, subsequent occurrence corrected to non-abbreviated term.
Line 170; write the meaning of NTR - done.
Line 387; although it appears later, writing the meaning of SEM in this his first writing - done.
Line 430 and 431; write in italics the name of N. benthamiana and N. tabacum -
interestingly, this mistake has occurred during pdf generation by the MDPI only system. It had been correct in the word file; we will take care for correctness upon proof correction.
Line 446; in figure 2D, for CP:TVCV it is indicated that it is wt, however in TMV it is not indicated - the image has been edited accordingly.
Line 519; since it appears here for the first time, indicate here the meaning of SPP - done.
Reviewer 2 Report
The results of this study demonstrate the wide possibilities of using turnip vein clearing virus (tobamovirus) particles coated with Staphylococcus aureus protein A (PA) domain as an immunosorbent. As part of the study, the authors proposed an approach for the isolation and purification of viral particles, that improved the degree of purification and preserved the integrity of the particles. The results of the study have promising prospects in the field of development for various areas of biodetection systems.
It seems to me that this is a holistic, well-written, and perfectly illustrated work that can be published in the journal Viruses.
However, I have a few comments that can improve the text.
1. Part of the Introduction section (lines 104-155) in my opinion is not directly related to this work. I suggest that the authors move this text to the Discussion section and connect it to the results of the study.
2. The result of the immunogold study (Figure 4C) confused. The figure shows selected and cropped micrographs, where colloidal gold is located next to the rod-shaped particles. Also alarming is the fact that no more than two gold nanoparticles bind to one viral particle, which is poorly consistent with other results. This result should be discussed in detail in the article. Authors may also include original uncut micrographs in the supplementary files.
3. In Figure 6, it is possible to clarify the description of sample No. 2 (control), since samples No. 1 and 2 are signed the same way (TVCV PA).
4. The list of references should be presented in a single format, including DOI links where necessary.
Author Response
We are pleased about this positive and very thorough evaluation of our manuscript, including comprehensible suggestions. We have addressed three out of the four carefully, with a single exception (1.) for which we have explained the rationale below, hoping that this seems reasonable to you and the Editor. Otherwise, we will change this in the course of a second revision timely. Many thanks!
Specific responses:
- Part of the Introduction section (lines 104-155) in my opinion is not directly related to this work. I suggest that the authors move this text to the Discussion section and connect it to the results of the study.-
We have cogitated about the best position of these data ourselves and were indeed of different opinions. However, the majority of people involved in the manuscript structure rated the information fundamental and introductory. Most of the items described in these paragraphs have not been under re-investigation in our work (i.e., taxonomic classification, genetic organization, OAs position, host plants, CP structure and subdomain sizes, positions of lysines) so that only the relevant ones for the study ( i.e., pI and RNA/particle lengths) have been taken up again in the discussion. Thus we left this section and Fig. 1 at the end of the introduction, because here the basic data on TVCV and TVCVPA are most easily found by readers generally interested in this tobamovirus and its engineered variants, but with no specific attention towards its biodetection potential. Such readers may be a considerably portion, as the Special Issue "Tobamoviruses 2023" most likely attracts the interest primarily of plant virologists. Furthermore, the discussion is quite extensive already in its current state. - The result of the immunogold study (Figure 4C) confused. The figure shows selected and cropped micrographs, where colloidal gold is located next to the rod-shaped particles. Also alarming is the fact that no more than two gold nanoparticles bind to one viral particle, which is poorly consistent with other results. This result should be discussed in detail in the article. Authors may also include original uncut micrographs in the supplementary files. -
This is an important comment and has been considered thoroughly. We have now included an overview micrograph even in the main article (new Fig. 4, replacing a cropped blow-up image, with the legend adapted) and have discussed the issue. It should be noted here (as not stated clearly enough in the original text and amended now) that the gold nanoparticle conjugates were directly bound to the PA sheath on the virus and not via indirect immunodetection of a primary antibody by a secondary gold IgG conjugate. This has been added explicitly to the results section. In addition, we have (unfortunately) used a relatively old gold-IgG preparation with most likely substantial amounts of IgGs having lost their gold load due to long-term storage, as we did not want to quantify the results. This limitation is now added to the methods section. Therefore, most likely two reasons account for the low gold NP decoration simultaneously, as explained in a new paragraph added to the discussion now on page 23 ("In initial proof-of-concept tests using IgG conjugates with gold nanoparticles of 15 nm diameter, only low decoration densities were obtained (Fig. 4). In addition to putative steric constraints, though, the presence of non-conjugated IgGs in the preparation was not excluded and could thus have contributed to the inefficient labeling (see methods section). ") [This is followed by a more clear discussion of the enzyme-conjugate binding results.] - In Figure 6, it is possible to clarify the description of sample No. 2 (control), since samples No. 1 and 2 are signed the same way (TVCV PA).
We agree with this suggestion and have modified Fig. 6 accordingly. - The list of references should be presented in a single format, including DOI links where necessary.
We apologize for having submitted a not yet uniform reference list; this has been amended.
Reviewer 3 Report
Minor remarks:
Section 2.1.1, Lines 173-175: What were the working concentrations of the four antibiotics used? Line 177: “Infiltrated by syringe”. Apparently, a syringe without a needle?
Section 3.1.1, Lines 542-543, also 893. Does the term “immature viruses/virions” imply replication-defective mutants that can be trans-complemented by helper virus? For tobamoviruses, this plot is relatively well studied.
Section 4. Line 894. Claim about “host RNA” that can be packaged into viral particles is supported by a single reference [112] from 1966. To my knowledge, all successful experiments on tobamovirus-like particles with the participation of foreign backbone RNA required the use of the viral origin of assembly.
Some editorial comments:
Line 554: “SDS-PAGE” is preferable than “SDS-PA”.
Line 629: “Catalytic activities of these goat IgGs” - obviously, should be IgG-conjugate.
Author Response
We are very grateful for these encouraging, favourable comments, the detailed inspection of our manuscript and the scientific questions, which we hope to have considered appropriately in the amended version. Please also refer to our explanations below. Thanks a lot!
Specific responses:
Minor remarks:
Section 2.1.1, Lines 173-175: What were the working concentrations of the four antibiotics used? Line 177: “Infiltrated by syringe”. Apparently, a syringe without a needle?
All these missing data have been added.
Section 3.1.1, Lines 542-543, also 893. Does the term “immature viruses/virions” imply replication-defective mutants that can be trans-complemented by helper virus? For tobamoviruses, this plot is relatively well studied.
This is an important additional possibility and has been neglected in both results and discussion section - thank you for proposing it. This putative further origin has now been added to the results as follows: "… representing mainly immature, incompletely encapsidated, and broken viruses, as well as particles containing subgenomic and optionally also mutated viral, and in some cases host nucleic acids (see discussion)". It has then been discussed together with the subsequent issue 4. on page 22, from the previous line 893 on, with some representative references.
Section 4. Line 894. Claim about “host RNA” that can be packaged into viral particles is supported by a single reference [112] from 1966. To my knowledge, all successful experiments on tobamovirus-like particles with the participation of foreign backbone RNA required the use of the viral origin of assembly.
Our initial reference (originally 112, Francki 1966) served as exemplary, summarizing quotation. However, there are various studies from different labs that have identified host transcripts (mRNAs and also ribosomal RNAs) without TMV origin of assembly (OAs) encapsidated in TMV CP tubes from systemically infected plants (co-existing with the TMV RNA-containing particles). In a few cases, 'pseudo-origins' resembling the OAs tertiary structure in the RNA were identified, but this has, to our knowledge, not been analyzed in-depth. Some of these articles are referenced now, in combination with a better overview of the distinct TMV-like tube types detected with and without viral sequences inside, and a few further references. Among those are also two previous papers from our team that review or introduce even more such studies. In addition, recent work of Saunders et al. (2022) provides experimental evidence that the major function of the TMV RNA's OAs is to increase its local affinity for CP, resulting in a more homogeneous nucleation and tube growth than without OAs. The authors thus conclude that the OAs does not serve as specificity determinant, which has been mentioned in the revised version as well. We hope these improved arguments may strengthen the respective, amended passage now on page 22 in the discussion.
Some editorial comments:
Line 554: “SDS-PAGE” is preferable than “SDS-PA” - this has been changed.
Line 629: “Catalytic activities of these goat IgGs” - obviously, should be IgG-conjugate -
this has been corrected. Thank you again!